# Brain milieu induces early microglial maturation through the BAX-Notch axis

Fangying Zhao[1,5], Jiangyong He [ORCID][1,2,5], Jun Tang[1], Nianfei Cui[1], Yanyan Shi [ORCID][1], Zhifan Li[1], Shengnan Liu[1], Yazhou Wang [ORCID][3], Ming Ma[1], Congjian Zhao[4], Lingfei Luo[1]✉ & Li Li[1,2]✉

Microglia are derived from primitive myeloid cells and gain their early identity in the embryonic brains. However, the mechanism by which the brain milieu confers microglial maturation signature remains elusive. Here, we demonstrate that the $bax^{cq55}$ zebrafish and $Bax^{tm1Sjk}$ mouse embryos exhibit similarly defective early microglial maturation. BAX, a typical pro-apoptotic factor, is highly enriched in neuronal cells and regulates microglial maturation through both pro-apoptotic and non-apoptotic mechanisms. BAX regulates *dlb* via the CaMKII-CREB axis calcium-dependently in living neurons while ensuring the efficient Notch activation in the immigrated pre-microglia by apoptotic neurons. Notch signaling is conserved in supporting embryonic microglia maturation. Compromised microglial development occurred in the $Cx3cr1^{Cre/+}$ $Rbpj^{fl/fl}$ embryonic mice; however, microglia acquire their appropriate signature when incubated with DLL3 in vitro. Thus, our findings elucidate a BAX-CaMKII-CREB-Notch network triggered by the neuronal milieu in microglial development, which may provide innovative insights for targeting microglia in neuronal disorder treatment.

As resident macrophages in the central nervous system (CNS), microglia guard and maintain CNS homeostasis by classic phagocytic performances[1]. Microglia are implicated in neural development and function[2]. They facilitate the integration of new neurons into neuronal circuits and release neurotrophic factors, including neural growth factor (NGF)[3,4] and brain-derived neurotrophic factor (BDNF)[5], to support the proliferation of neural progenitor cells. Microglia sculpt the synapses through the complement system, which is composed of complement receptor 3 (CR3), located on microglia, and C3 expressed by neurons, in mediating the forgetting and erasure of stored memories from specific neuronal groups[6]. Anticipatively, microglia are intensively involved in the progression of neurological diseases and are a prospective therapeutic target.

Unlike other CNS cell types, which arise from the neuroectoderm, microglia derive from primitive myeloid cells of mesoderm origins[7]. They enter the embryonic brain via default pathways from around E9.5-E10 and 22–40 h post-fertilization (hpf) in mouse and zebrafish, respectively, until the formation of the blood-brain barrier[8–10]. A large number of neurons provide the signals to attract and locate microglial cells in the brains. For example, some apoptotic neurons induce microglial brain colonization by releasing "find me" signals, such as lysophosphatidylcholine or adenosine 5′-triphosphate (ATP)[11,12]. However, interleukin (IL) 34, a colony-stimulating factor 1 receptor (CSF1R) ligand expressed in neurons, functions independently in controlling microglial immigration[13]. In embryonic mouse brains, microglial progenitors express some genes associated with neural maturation and

[1]Institute of Developmental Biology and Regenerative Medicine, Key Laboratory of Freshwater Fish Reproduction and Development, Ministry of Education, Southwest University, 400715 Chongqing, P.R. China. [2]Research Center of Stem Cells and Ageing, Chongqing Institute of Green and Intelligent Technology, Chinese Academy of Sciences, 400714 Chongqing, P.R. China. [3]Department of Neurobiology and Institute of Neurosciences, School of Basic Medicine, Fourth Military Medical University, 169 Chang Le Xi Road, 710032 Xi'an, Shaanxi, P.R. China. [4]Chongqing Engineering Research Center of Medical Electronics and Information Technology, School of Bioinformatics, Chongqing University of Posts and Telecommunications, 400065 Chongqing, P.R. China. [5]These authors contributed equally: Fangying Zhao, Jiangyong He. ✉e-mail: lluo@swu.edu.cn; lili@cigit.ac.cn

synaptic pruning since E14 and mature in a few postnatal weeks[14]. This process is accomplished through complex interactions with neurons, astrocytes, and other cell types[15–17]. However, the mechanism by which the brain milieu confers a specific microglia signature, particularly in the embryonic stages, remains elusive. Although transforming growth factor beta (TGF-β) and CSF1, which are mainly provided by astrocytes and neurons, have been documented to be indispensable in promoting certain microglia-specific gene expression in the postnatal stages[18,19], the microglia signatures are rapidly lost in culture media containing CSF-1, TGF-β2, and cholesterol[20]. These data indicate that additional CNS-specific cues are required in microglial maturation. The optical transparency and considerable genetic screening advantages of zebrafish enable the discovery of additional factors involved in the conserved microglial development, particularly in their maturation during the embryonic stages. Upon settling in the brain, embryonic zebrafish macrophages undergo a phenotypic transition into early pre-microglia in the cephalic mesenchyme at 55–60 hpf[10]. These cells differentiate and mature rapidly to obtain the core microglia signature, with strong upregulation of specific genes, such as *apoeb*, and increasingly phagocytose cell debris from 3 days post-fertilization (dpf) onwards[10,12].

Notch signaling participates in many cell–cell communication events via variable ligand-receptor pairing[21], and is involved in microglia polarization and activation[22,23]. In mice, Notch1 knockdown causes a significant reduction in the number of activated microglia in the ipsilateral ischemic cortex and a corresponding decreased expression of pro-inflammatory cytokines[24]. In the BV-2 microglial cell line, Notch signaling is detected and involved in cell migration and morphological transformation upon stimulation[25]. However, the mechanism by which the brain microenvironment deploys Notch signaling to dictate embryonic microglial development is poorly understood. BCL2-associated X protein (BAX), a typical pro-apoptotic factor, participates in the efficient flux of $Ca^{2+}$ between the cytosol and the endoplasmic reticulum (ER) to control neuronal $Ca^{2+}$ homeostasis[26,27]. The genetic deletion of BAX in mice leads to cerebellar neurogenesis disruption, medulloblastoma formation, and a significantly reduced neuronal $Ca^{2+}$ flux during the N-methyl-D-aspartate (NMDA) excitation period[26,28]. However, the roles of BAX in microglial development are poorly elucidated, although BAX-dependent neuronal apoptosis affects the disease-associated microglia (DAM)-related gene expression in postnatal retinal microglia in mice[27].

The zebrafish (*Danio rerio*) *baxa* is homologous to the mammalian *BAX* and is conserved in apoptotic regulation[29]. In the present study, we identify a *bax* mutant zebrafish allele that compromises microglial maturation and reduces neuronal apoptosis in the embryonic stages. Combined with the results obtained using *Bax^tm1Sjk* mice, our data reveal that BAX is highly enriched in neuronal cells and dictates the early microglia signature via Notch signaling. BAX regulates a specific neuronal activity to activate the calcium-calmodulin-dependent protein kinase II (CaMKII)-cAMP-response element-binding protein (CREB) axis, which controls *dlb* expression, to provide a Notch ligand in living neurons. BAX simultaneously provides attraction signals through the purinergic signaling of apoptotic neurons to positionally retain pre-microglia for an effective Notch signaling transmission. Notch signaling is conserved in early microglial maturation regulation, according to the compromised microglia signatures observed in embryonic *Cx3cr1^Cre/+Rbpj^fl/fl* mice brains and the apparent presentation of microglia characteristics in DLL3-containing in vitro cultures.

## Results

### Identification of a *bax^cq55* mutant zebrafish with a compromised microglial maturation in the embryonic stages

We imaged the process of embryonic microglial maturation in a *Tg(apoeb:GFP;coro1a:DsRed)* midbrain, in which *apoeb* labels microglia[30] and *coro1a* indicates myeloid cells[31]. Approximately three long thin apoeb-GFP⁻coro1a-DsRed⁺ (a-G⁻c-D⁺) pre-microglia (PM) per

midbrain with limited branches initially appeared at 2 dpf, which actively moved at a speed of 1.47 ± 0.20 μm/min and transiently touched 3–4 HuC-GFP⁺ (H-G⁺) neuronal cells (Fig. 1a-g, Supplementary Fig. 1a, b). The a-G⁻c-D⁺ PM rarely engulfed annexin V⁺/active Casp3⁺ signals, neutral red (NR)⁺ particles and DsRed⁺ bacteria (*E. coli*) (Fig. 1h, Supplementary Fig. 1c-g, Supplementary Movie 1). One day later, approximately 25 a-G⁻c-D⁺ cells per midbrain expressed *apoeb*, *p2ry12* and *p2ry6* and differentiated to apoeb-GFP⁺coro1a-DsRed⁺ (referred to as a-G⁺c-D⁺) cells (Fig. 1a–c, Supplementary Fig. 1a, b, h, i, j, Supplementary Movie 2). Compared to a-G⁻c-D⁺ counterparts, a-G⁺c-D⁺ cells were morphologically amoeboid and were thus named amoeboid microglia (AM). The size and mobility of a-G⁺c-D⁺ AM, which contacted much more with H-G⁺ cells, increased to 2,416.00 ± 178.50 μm³ and significantly decreased to 0.21 ± 0.04 μm/min, respectively (Fig. 1b, d–g). The phagocytic abilities of a-G⁺c-D⁺ AM were robustly enhanced to collect apoptotic cells, NR⁺ particles, and DsRed⁺ bacteria, and these cells expressed higher transcript levels of phagocytic-related genes (Fig. 1h, Supplementary Fig. 1c–g). Notably, zebrafish microglia could be derived from the a-G⁻c-D⁺ PM, which matured into a-G⁺c-D⁺ AM (Fig. 1a, b). Then, large portions of AM transformed into ramified microglia (RM) (86.21% at 6 dpf) and inhabited the brain (Fig. 1a–c).

Based on the co-staining results obtained using *apoeb* with *p2ry12*[32] and *p2ry6*[33] (Supplementary Fig. 1i) and on previous reports[10,34], we utilized whole-mount in situ hybridization (WISH) of *apoeb* to indicate the matured microglia in a forward genetic screening of an N-ethyl-N-nitrosourea (ENU)-treated zebrafish library[35], to identify the instructive molecules in microglial maturation in the embryonic stages. A total of 203 crossing pairs were examined. The Chongqing number 55 pairs (*cq55*) were derived from the embryos exhibiting few *apoeb*⁺ signals and were maintained as a candidate mutant allele. The complementation tests indicated that the *cq55* mutant was neither caused by *csf1ra*[10], *pu.1*[36], nor *irf8*[37] and that they warranted further investigation. The careful characterization indicated that the *cq55* mutant initially gave rise to comparable 3-4 a-G⁻c-D⁺ cells at 2 dpf (Fig. 1a, c, Supplementary Fig. 1a-b). Compared with that in siblings, the mutant a-G⁻c-D⁺ cells (*cq55* M) failed in expressing the microglia signature genes, *apoeb*, *slc7a7*[38], *p2ry12*, and *p2ry6*, during embryonic development (Fig. 1a, Supplementary Fig. 1a, h). These *cq55* M recapitulated the identical morphological appearance, closed transcript signature, similar movement and contact behaviors, and limited phagocytic abilities of PM (Fig. 1a, d-h, Supplementary Fig. 1e-g, k, Supplementary Movie 1). Additionally, *cq55* M was not confined to the midbrain (Supplementary Movie 1). Over half of the cells entered the eyes, hindbrains, and ventral brains, causing a significant reduction in the c-D⁺ microglia pools (Fig. 1i, j). These data indicate compromised microglial recruitment and/or maturation in *cq55* embryos (Fig. 1k).

Genetic recovery analysis indicated that the mutation was located in linkage group 3 (LG3) (Supplementary Fig. 2a). A T-to-A mutation in the *baxa* gene (ENSDARG00000020623), the homolog of *BAX*, was identified (Supplementary Fig. 2a, b). The mutation caused an Ile to Asn substitution in the conserved BH1 domain and significantly disrupted the BAX protein levels (Supplementary Fig. 2b–d). Two newly generated *bax* null alleles with 1- and 58-bp deletions exhibited identical microglia deficiency phenotypes (Supplementary Fig. 2b, d, e). Widespread supplementation of WT, but not mutant *bax* mRNA, or a controllable supply of *bax* in heat-shock-induced *Tg(hsp70:bax;-cryaa:Cerulean)* in the *bax^cq55* mutants significantly induced *bax* transcription and recovered the *apoeb*⁺ and NR⁺ signals (Supplementary Fig. 2f–j).

### *Bax* is expressed in neurons and non-cell-autonomously regulates embryonic microglial maturation

*Bax* transcripts were clearly detected in the brain, eyes, and digestive organs of zebrafish embryos using WISH (Fig. 2a). In the brain, *bax* was much more enriched in the Sox2⁺ and H-G⁺ cells than in the c-D⁺ cells

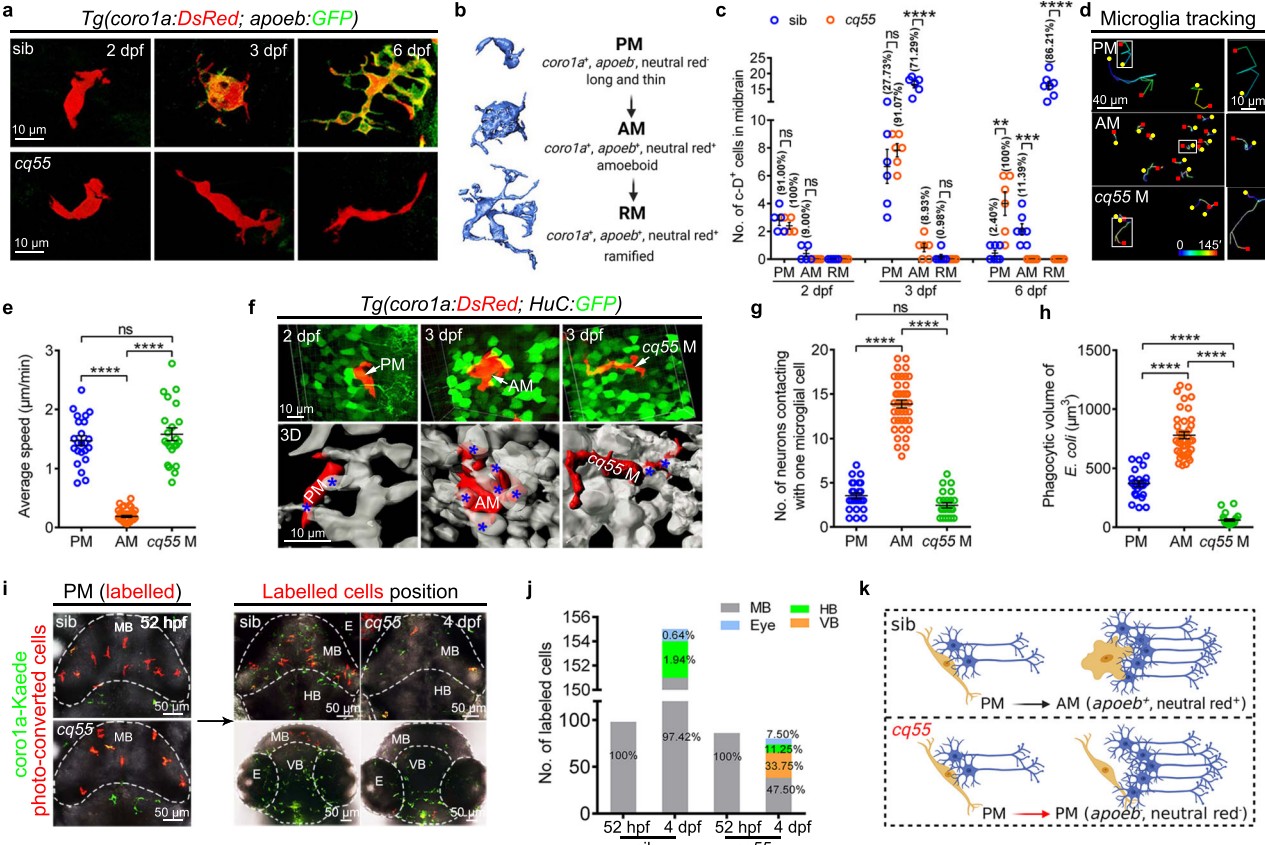

**Fig. 1 | Microglial differentiation in a larval zebrafish midbrain. a** Representative images of midbrain microglial cells. **b** Zebrafish microglial development process. **c** The number and proportion of PM, AM and RM in midbrains (2 dpf: sib and *cq55*: $n = 5$; 3 dpf: sib and *cq55*: $n = 6$; 6 dpf: sib: $n = 7$, *cq55*: $n = 6$). Each dot denotes one fish. **d** The movement trajectory of c-D[+] cells during 145 min. Each line represents one cell. The right panels are the enlarged images of boxed cells. The yellow circle and red square indicates start and end points, respectively. **e** The movement speed of c-D[+] cells in **d** ($n = 23$ PM, 41 AM and 22 *cq55* M from 6 fish per group). **f** Representative images (top) and 3D reconstructions (bottom) of c-D[+] PM, AM or *cq55* M (red) contacting HuC-GFP[+] neurons (green and white colors in top and bottom panels). White arrows and blue asterisks mark the c-D[+] cells and touched neurons. **g** The numbers of H-G[+] neurons contacting PM, AM or *cq55* M ($n = 24$ PM

from 6 fish; $n = 40$ AM from 5 fish; $n = 24$ *cq55* M from 6 fish). **h** The volume of phagocytized *E. coli* ($n = 23$ PM from 6 fish; $n = 40$ AM from 5 fish; $n = 23$ *cq55* M from 6 fish). **i** Imagings of c-K[+] PM (left panel). Coronal (right top) and transverse (right bottom) views of red c-K[+] cells at 4 dpf. **j** The number and proportion of red c-K[+] cells in different regions in (**i**) ($n = 10$ per group). **k** Schematic overview of microglia phenotypes, created with BioRender.com. sib siblings, M microglia, PM pre-microglia, AM amoeboid microglia, RM ramified microglia, c-D coro1a-DsRed, c-K coro1a-Kaede, MB midbrain, HB hindbrain, VB ventral brain, E eye. The numbers (**c**) and (**j**) indicate the average percentage. Each dot in (**e**), (**g**), and (**h**) represents one cell. Error bars, mean ± SEM. **$P < 0.01$; ***$P < 0.001$; ****$P < 0.0001$; ns no significant, Unpaired two-tailed Student's *t* test. Source data are provided as a Source Data file.

(Fig. 2b-e). Transplantation assays and transgenic rescue strategies were conducted to explore the mechanisms by which *bax* regulates microglial development. The H-G[+] cells were sorted from 55 hpf WT or *bax^cq55*/*Tg(HuC:GFP)* brains and transplanted into the midbrains of recipient *bax^cq55* mutant embryos (Fig. 2f). Consequently, the transplantation of isolated WT rather than mutant H-G[+] cells into the *bax^cq55* mutant brains led to a reappearance of *apoeb*[+] signals at 17 h post-transplantation (hpt, Fig. 2g, h). Furthermore, *Tg(NBT:bax)* or *Tg(coro1a:bax)* were generated, in which the *bax* coding sequence was controlled by the neuronal cell-specific (*NBT*) or macrophages/microglia (*coro1a*) regulatory elements. The complete recovery of *apoeb*[+] and NR[+] phenotypes and an increase in the AM and RM populations were observed in *bax^cq55*/*Tg(NBT:bax)* (Fig. 2i, j). However, no rescue effects were discernable in *bax^cq55*/*Tg(coro1a:bax)* (Fig. 2i, j). These results collectively indicate that neuronal BAX non-cell-autonomously regulates embryonic microglial maturation (Fig. 2k).

## Classical pro-apoptotic effects of BAX are required for the brain residency of PM

BAX is a canonical pro-apoptotic factor[27]. The counts of active Casp3[+] and acridine orange[+] (AO[+]) signals were significantly reduced;

however, the HuC-GFP[+] pool expanded obviously in the *bax^cq55* mutants compared with that in their WT counterparts (Supplementary Fig. 3a−c). The AO[+] signals were effectively recovered upon *bax* replenishment in the *bax^cq55*/*Tg(hsp70:bax)* (Supplementary Fig. 3b). Apoptotic neurons have a crucial function in microglial development[11,12]. To weigh the pro-apoptotic roles of BAX in this scenario, we generated a *Tg(NBT:DenNTR)* line, in which the fluorescent protein Dendra2 was fused to NTR, enabling the easy visualization of NBT[+] cells. The application of MTZ to *Tg(NBT:DenNTR)* led to increased active Casp3[+] signals and reduced NBT[+] pools relative to those in the DMSO controls (Fig. 3a). This is because NTR catalyzed the conversion of a nontoxic prodrug into a cytotoxic agent in the NBT[+] cells[39]. Consequently, a significant increment of *mpeg1*[+] and c-D[+] cells was observed in the MTZ- rather than in the DMSO-treated WT and *bax^cq55* mutants bearing the *Tg(NBT:DenNTR)* background (Fig. 3a, b). Counterintuitively, no *apoeb*[+] and NR[+] signals were simultaneously detected in *bax^cq55* mutant brains despite the higher intensity of the *apoeb*[+] and NR[+] signals in the MTZ-treated WT than in the controls (Fig. 3a). Similarly, the application of PAC-1, a procaspase-activating compound, induced obvious AO[+] signals in the *bax^cq55* mutant brains (Supplementary Fig. 3d). No *apoeb*[+] signals presented again in the *bax^cq55*

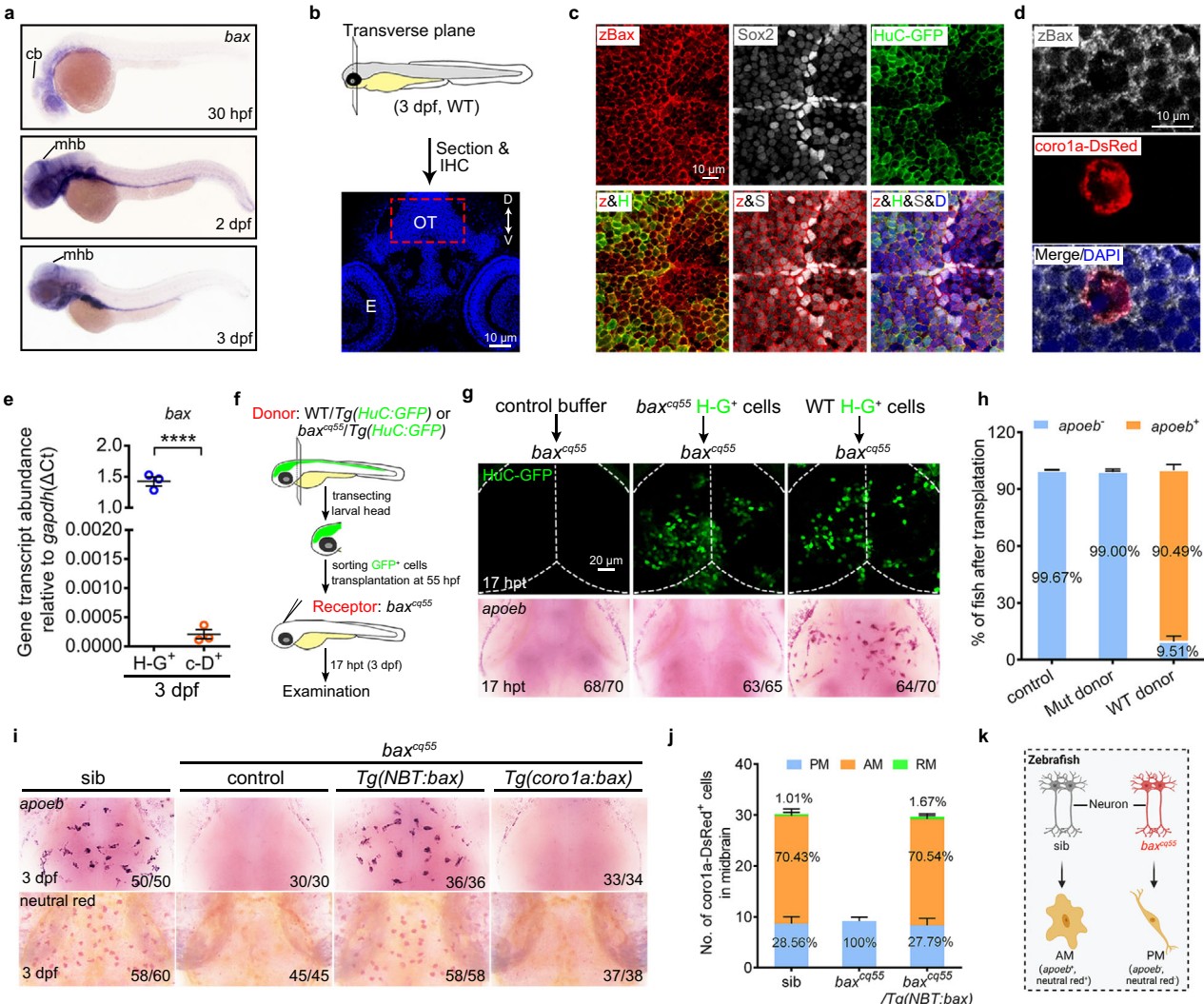

**Fig. 2 | Bax is expressed in neurons and non-cell-autonomously regulates microglial maturation. a** WISH of *bax* in 30 hpf, 2 dpf, and 3 dpf zebrafish embryos. **b** Schematic illustration of the anatomical position and observational region (red square) in (**c**) and (**d**), created with photoshop CS6. **c**, **d** Immunofluorescence of zBax (z) with Sox2 (S) and HuC-GFP (H) (**c**) or coro1a-Dsred (**d**) in 3 dpf OT region. **e** qPCR results of the *bax* transcriptional levels in the sorted H-G⁺ and c-D⁺ cells from 3dpf WT midbrains. The data are from three independent experiments. Each dot represents an independent experiment. **f** The transplantation workflow created with photoshop CS6. **g** Fluorescent images of H-G⁺ neurons (top) and WISH of *apoeb* (bottom) in *bax^cq55* recipient midbrains at 17 hpt. **h** The percentage of *bax^cq55* recipients showing rescued *apoeb⁺* signals in (**g**). The number in each histogram indicates the average percentage. Data are pooled

from three independent experiments. **i** Representative images of *apoeb* WISH and NR staining signals in the midbrains of siblings, *bax^cq55*, *bax^cq55/Tg(NBT:bax)* and *bax^cq55/Tg(coro1a:bax)*. **j** The number and percentage of PM, AM and RM in 3 dpf midbrains (*n* = 6 fish in each group). **k** Schematic diagram of neuronal Bax regulating microglial maturation, created with BioRender.com. cb cerebellum, mhb mid-hindbrain boundary, IHC immunohistochemistry, D dorsal, V ventral, E eye, OT optic tectum, zBax zebrafish Bax, hpt hours post transplantation, WISH wholemount in situ hybridization, NR neutral red. The number in each histogram of (**h**) and (**j**) indicates the average percentage. Numbers in the right corners in (**g**) and (**i**) indicate the counts of embryos with a typical appearance (first number) in the total examined fishes (last number). Error bars, mean ± SEM. ****P < 0.0001, Unpaired two-tailed Student's *t* test. Source data are provided as a Source Data file.

mutants despite the amoeboid appearance of c-D⁺ cells (Supplementary Fig. 3d). These data indicated that the pro-apoptotic role of *bax* is crucial for the brain location of *mpeg1⁺* and c-D⁺ PM; nonetheless, accomplishing their maturation in the *bax^cq55* mutants was insufficient. To provide more evidence, we focused on the nucleotides, such as ATP, employed by dead neurons to attract and hold microglial cells[12,32]. We injected apyrase, an ATP-hydrolyzing enzyme, directly into the midbrain of 3 dpf WT embryos. The c-D⁺ cells actively moved away, causing a marked decrement in the coro1a-Kaede⁺ (c-K⁺) and *apoeb⁺* populations in the treated brain (Supplementary Fig. 3e, Supplementary Movie 3). Similarly, the injection of probenecid and carbenoxolone (CBX), the pannexin-1 (panx1) gap junction inhibitors that block ATP release, caused *apoeb⁺* cells to robustly vanish (Supplementary Fig. 3f). These phenotypes mirrored those observed in the *bax^cq55*

mutants. We then supplied ATP to the *bax^cq55/Tg(coro1a:Kaede)* brains. Despite a significant accretion of c-K⁺ cells, these cells acquired *apoeb⁺* signals to a limited extent (Supplementary Fig. 3g). These results collectively indicate that solely increasing the apoptotic cell number or signals in *bax^cq55* mutant brains failed to accomplish microglial maturation.

## BAX promotes embryonic microglial maturation via Notch signaling

To identify the additional molecules induced by BAX in embryonic microglial maturation in the *bax^cq55* mutants, H-G⁺ cells (neurons) and photo-conversion labeled red c-K⁺ cells (microglial cells) were collected from 3 dpf siblings and *bax^cq55* mutant brains to perform parallel RNA sequencing (RNA-seq). The data revealed that a total of 783 genes

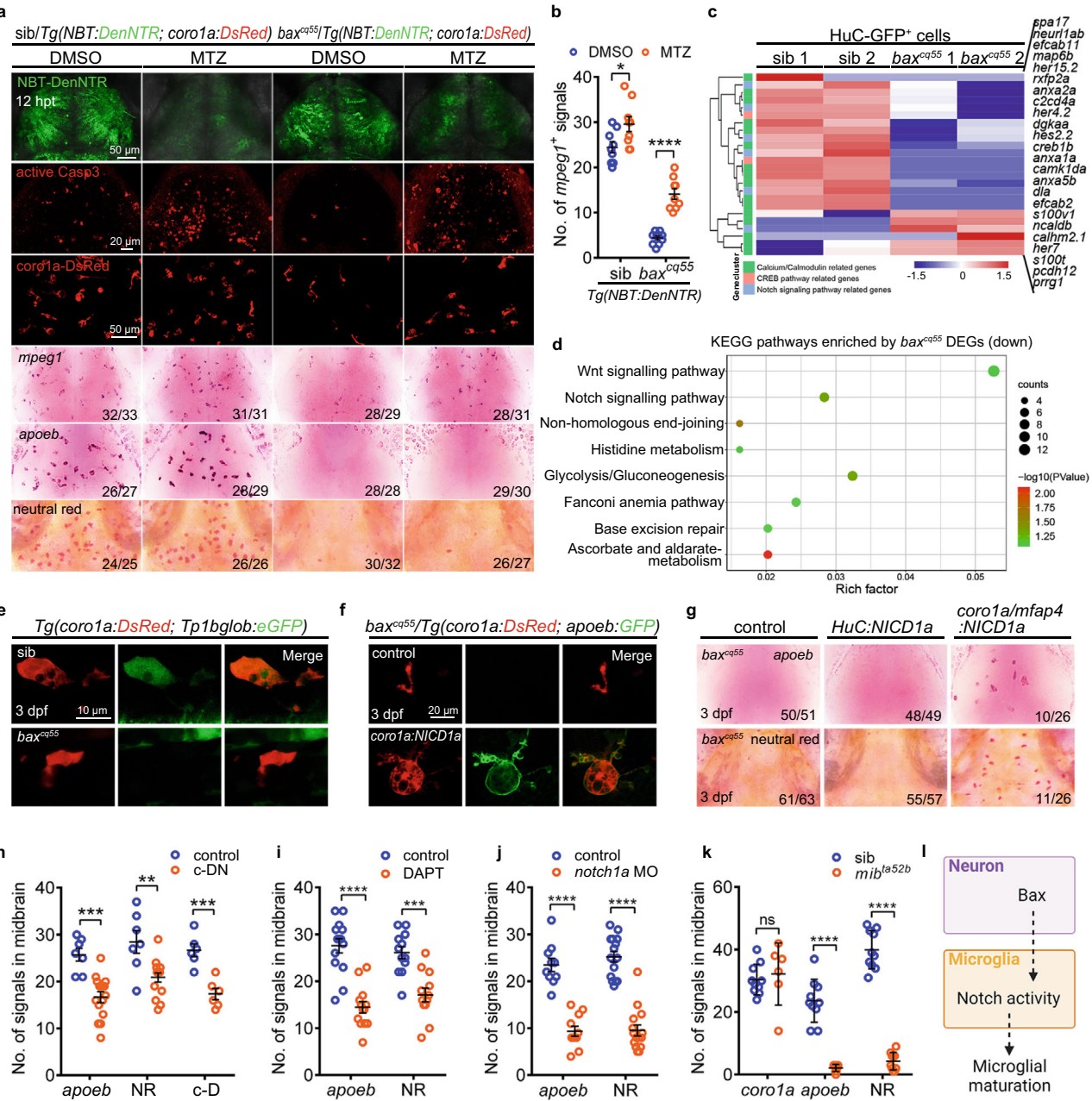

**Fig. 3 | Bax controls microglial maturation via Notch signaling. a** The images of NBT-DenNTR[+], active Casp3[+], coro1a-DsRed[+], *mpeg1* WISH, *apoeb* WISH and NR staining signals in DMSO and/or MTZ-treated midbrains at 12 h post-treatment (hpt). **b** Quantification of *mpeg1*[+] signals in (**a**) (DMSO: sib: $n = 9$, *bax^cq55*: $n = 10$; MTZ: sib: $n = 9$, *bax^cq55*: $n = 9$). **c** Heat-map of altered genes in 3 dpf H-G[+] cells. The altered Calcium/Calmodulin, CREB, and Notch signaling pathway were indicated by different colors. **d** KEGG pathways of the down-regulated genes in 3 dpf red c-K[+] cells. Enrichment *P* values were calculated with Fisher's Exact test. Multiple testing was corrected using FDR correction. **e** Representative images of DsRed[+] and GFP[+] signals. **f** Representative images of a-G[+] signals after supplying *coro1a:NICD1a*. **g** Representative images of *apoeb* WISH and NR staining after supplying *HuC:NICD1a* or *coro1a/mfap4:NICD1a*. **h** Quantification of *apoeb*[+], NR[+] and c-D[+] signals in 3 dpf midbrains of *Tg(coro1a:DN-MAML-FLAG)* (c-DN) (*apoeb*: control: $n = 7$, c-DN:

$n = 15$; NR: control: $n = 7$, c-DN: $n = 15$; c-D: $n = 6$ per group). **i, j** Quantification of *apoeb*[+] and NR[+] signals in 3 dpf midbrains treated with DAPT (**i**, *apoeb*: control: $n = 14$, DAPT: $n = 13$; NR: $n = 12$ per group) or *notch1a* MO (**j**, *apoeb*: control: $n = 10$, *notch1a* MO: $n = 11$; NR: $n = 15$ per group). **k** Quantification of *coro1a*[+], *apoeb*[+] and NR[+] signals in 3 dpf midbrains (*coro1a*: sib: $n = 10$, *mib^ta52b*: $n = 6$; *apoeb*: sib: $n = 10$, *mib^ta52b*: $n = 9$; NR: $n = 10$ per group). **l** A diagram of neuronal Bax dictating microglial development via Notch signaling, created with BioRender.com. Each dot in (**b**, **h–k**) denotes one fish. Numbers in the right corners in (**a**) and (**g**) indicate the counts of embryos with a typical appearance (first number) in the total examined fishes (last number). Error bars, mean ± SEM. *$P < 0.05$; **$P < 0.01$; ***$P < 0.001$; ****$P < 0.0001$; ns no significant, Unpaired two-tailed Student's *t* test. Source data are provided as a Source Data file.

presented significant transcriptional differences between the sibling and *bax^cq55* mutant H-G[+] cells, and 1,112 genes exhibited transcriptional differences in c-K[+] cells (Supplementary Fig. 4a, b). Differences were also observed in a set of biological processes between the *bax^cq55* mutant H-G[+] cells and the controls (Supplementary Fig. 4c). Notably,

the levels of the Notch signaling factors concomitantly exhibited pronounced disparities in both *bax^cq55* mutant H-G[+] and c-K[+] cells compared to the corresponding controls (Fig. 3c, d). Concordantly, the imaging of the Notch activity in the *Tg(coro1a:DsRed;Tp1bglob:eGFP)* and/or *Tg(apoeb:GFP;Tp1bglob:hmgb1-mCherry)* reporter lines

revealed that the GFP⁺ and/or mCherry⁺ signals were clearly observed in 3 dpf c-D⁺ and/or a-G⁺ cells (Fig. 3e, Supplementary Fig. 4d). However, these signals were sparingly detected in the *bax^cqSS* mutant counterparts (Fig. 3e). The western blot results of a validated Notch1 antibody indicated a significant decline in the NICD1 level in the *bax^cqSS* mutants compared to that in their siblings (Supplementary Fig. 4e). These data indicate that the Notch signaling, which might be indispensable in microglial development, was compromised in *bax^cqSS* mutants. To this end, the rescue experiments were conducted by supplying the functional Notch1a intracellular domain (NICD1a) to macrophages/microglia (*coro1a/mfap4:NICD1a*) and neurons (*HuC:NICD1a*). As expected, the transient supplementation of *coro1a/mfap4:NICD1a*, but not *HuC:NICD1a*, to the *bax^cqSS* mutants effectively promoted the reappearance of *apoeb⁺* signals in c-D⁺ populations in approximately 40% of the treated embryos (Fig. 3f, g). These restored c-D⁺ cells phagocytosed the dead cells and NR⁺ dyes (Fig. 3f, g).

## Notch signaling is indispensable in microglial development

To validate the roles of Notch in microglial development, we isolated 24,742 coro1a-DsRed⁺ (c-D⁺) cells from 4 dpf zebrafish brains to conduct single-cell RNA sequencing (scRNA-seq, Supplementary Fig. 4f). Two subtypes of microglia signatures were annotated using the *t*-distributed stochastic neighbor embedding (*t*SNE) method (Supplementary Fig. 4g, h). The microglia-specific genes (*apoeb*, *p2ry12*, and *slc7a7*) were intensively detected in cluster 2 (Supplementary Fig. 4i), which presented mature microglia signatures including immune response (GO:0006955), phagocytosis (GO:0006909), microglia/macrophage differentiation (GO:0030225, GO:0014004) (Supplementary Fig. 4j). However, the cells of cluster 1 presented a high expression of *epd*, *atp5mc1*, *calm1a*, and *hmgb1a*, which are associated with an active metabolism and translation, including the isocitrate metabolic process (GO:0006102), cytoplasmic translation (GO:0002181), and translational elongation (GO:0006414) (Supplementary Fig. 4i, j). *notch1a*, *notch2*, and *her9* were significantly detected in both clusters, particularly in cluster 2 (Supplementary Fig. 4i), implying the involvement of Notch in physical microglial development. To provide further evidence, we developed a *Tg(coro1a:DN-MAML-FLAG)* strain, in which a dominant-negative (DN) isoform of the murine mastermind-like (MAML) protein was fused to FLAG and controlled by the *coro1a* promoter to specifically inhibit Notch activities in microglia (referred to as c-DN). FLAG fluorescence was evident in the Lcp1⁺a-G⁺ cells (Supplementary Fig. 5a). The transcript levels of *her4.1*, a typical Notch target, were significantly reduced in the c-D⁺ cells of *Tg(coro1a:DN-MAML-FLAG)* compared with those in the controls (Supplementary Fig. 5b). Consequently, obvious suppression of *apoeb⁺*, NR⁺, and c-D⁺ cells were observed in *Tg(coro1a:DN-MAML-FLAG)* embryos compared with that in the controls (Fig. 3h, Supplementary Fig. 5c). Furthermore, the proportions of the AM and RM populations were simultaneously and significantly reduced (Supplementary Fig. 5d). Concordantly, the transient global inhibition of Notch signaling via interference with the γ-secretase inhibitor (DAPT) and *notch1a* morpholino oligos (MOs), or the heat-shocking *Tg(hsp70:DN-MAML-GFP)* line, led to a marked reduction in the *apoeb⁺* and NR⁺ pools (Fig. 3i, j, Supplementary Fig. 5e, f). *mib^taS2b*, a Notch signaling-depleted mutant, displayed a limited number of *apoeb⁺* and NR⁺ cells; however, the *coro1a⁺* population was comparable to that of its siblings (Fig. 3k, Supplementary Fig. 5g). The supplication of *coro1a:NICD1a* efficiently rescued the phenotypes of *apoeb⁺* and NR⁺ cells in *mib^taS2b* (Supplementary Fig. 5g). These data jointly demonstrate the integral roles of Notch activation by BAX in promoting early microglial maturation (Fig. 3l).

## Neuronal DeltaB is required for microglial Notch activation

To capture the neuronal ligands involved in triggering Notch activation in microglia, the expression profiles of *delta* and *jagged* members, including *dla*, *dlb*, *dlc*, *dld*, *jag1a*, *jag1b*, and *jag2b* were examined using WISH (Fig. 4a, Supplementary Fig. 5h). All of these ligands were apparent in the midbrain. However, the intensity of the *dlb* signals declined impressively in the *bax^cqSS* mutants compared to that of their siblings (Fig. 4a). We examined the cellular localization of DeltaB using a validated antibody of the mammalian homolog DLL3 (Supplementary Fig. 5i). The double staining results indicated that the DeltaB signals mainly appeared in the H-G⁺; however, they were sparingly detected in the c-D⁺ cells (Fig. 4b, c) and were neither presented in the TUNEL⁺ nor the engulfed cells (Fig. 4b, c). In the H-G⁺ cells, the DeltaB signals were merged with the β-catenin⁺ signals of the fringe membrane (Fig. 4d). However, their intensities and levels diminished significantly in the *bax^cqSS* mutants compared to those in their siblings (Fig. 4e, f). A *dlb* mutant allele that produced an unstable truncated form of DeltaB was created (Fig. 4g, Supplementary Fig. 5j). The *dlb⁻/⁻* mutants manifested a substantial absence of *apoeb⁺* and NR⁺ signals; nonetheless, they exhibited no evident alterations in the active Casp3⁺, *coro1a⁺*, and *mpeg1⁺* signals (Fig. 4h, i, Supplementary Fig. 5k). Similar results reappeared in *dlb* morphants (Fig. 4j, Supplementary Fig. 5l). Transiently providing *HuC:dlb* evidently prompted the recovery of *apoeb⁺* and NR⁺ cells in more than 29% of the *bax^cqSS* mutant embryos (Fig. 4k, l). However, these effects were significantly reduced to less than 13% upon suppressing the Notch activities in *coro1a⁺* microglial cells by additionally supplying *coro1a:DN-MAML-FLAG* to the *bax^cqSS* mutants (Fig. 4k, l). Notably, we observed that *HuC:dlb* supplementation promoted the advanced appearance of *apoeb⁺* and NR⁺ signals in 21.98% of the WT midbrains at 2 dpf, one day earlier than its physiological appearance. This effect was significantly increased to 31.20% after the synergistic application of ATP and *HuC:dlb* (Supplementary Fig. 5m, n). At the same time, ATP depletion decreased the efficiency to 10.77% (Supplementary Fig. 5m, n). These data indicate the important role of neuronal DeltaB in triggering Notch activity in embryonic microglial maturation upon their proper positioning via purinergic signaling (Fig. 4m).

## BAX regulates *dlb* via spontaneous neuronal activity

We then explored the molecular clues of BAX in neuronal *dlb* regulation by re-analyzing the RNA-seq profile. Transcriptional changes associated with several molecular pathways that are potentially engaged by neuronal activity, including the calcium/calmodulin-related pathways, G-protein coupled receptor, and protein kinase pathways, were observed between the *bax^cqSS* HuC-GFP⁺ neurons and the WT controls (Fig. 3c, Supplementary Fig. 4c). We monitored the spontaneous neural activities by imaging a *Tg(HuC:GCaMP6s)* larva, in which neurons express the genetically encoded calcium indicators GCaMP6s. Annexin V (AV) was administered to exclude apoptotic cells. The embryonic neuronal activity appeared from 2.5 dpf and was significantly detected at 3 dpf, as indicated by the neurons with calcium events quantified between 2–5 dpf in the unilateral midbrain (Supplementary Fig. 6a). Six WT and six *bax^cqSS* mutant embryos were stochastically selected to perform the imaging at 3 dpf. One side of each midbrain was serially captured for 47 min by using the same protocol. In total, 110 and 56 neurons were snapped in WT and *bax^cqSS* mutant embryos, respectively and they demonstrated different GCaMP6s⁺ transient fluorescence dynamics. Continual and strong intensities of the GCaMP6s⁺ signals were observed in the AV⁺ apoptotic neurons in the WT group; however, these signals were rare in the *bax^cqSS* mutants (Fig. 5a, b). In GCaMP6s⁺AV⁻ neurons, 1,195 calcium transients were identified in 99 WT neurons and presented at different frequencies. We primarily classified the calcium events into two types according to a GCaMP6s⁺ fluorescence transient of nine, which appeared at a limited level in the calcium events of frequent present groups (Supplementary Fig. 6b). The number of neurons with a high frequency (HF) of GCaMP6s⁺ transients (≥ nine times during the imaging period) was significantly reduced from 53, in the siblings, to 17, in the *bax^cqSS* mutants (Fig. 5a–d, Supplementary Movie 4). The calcium changes,

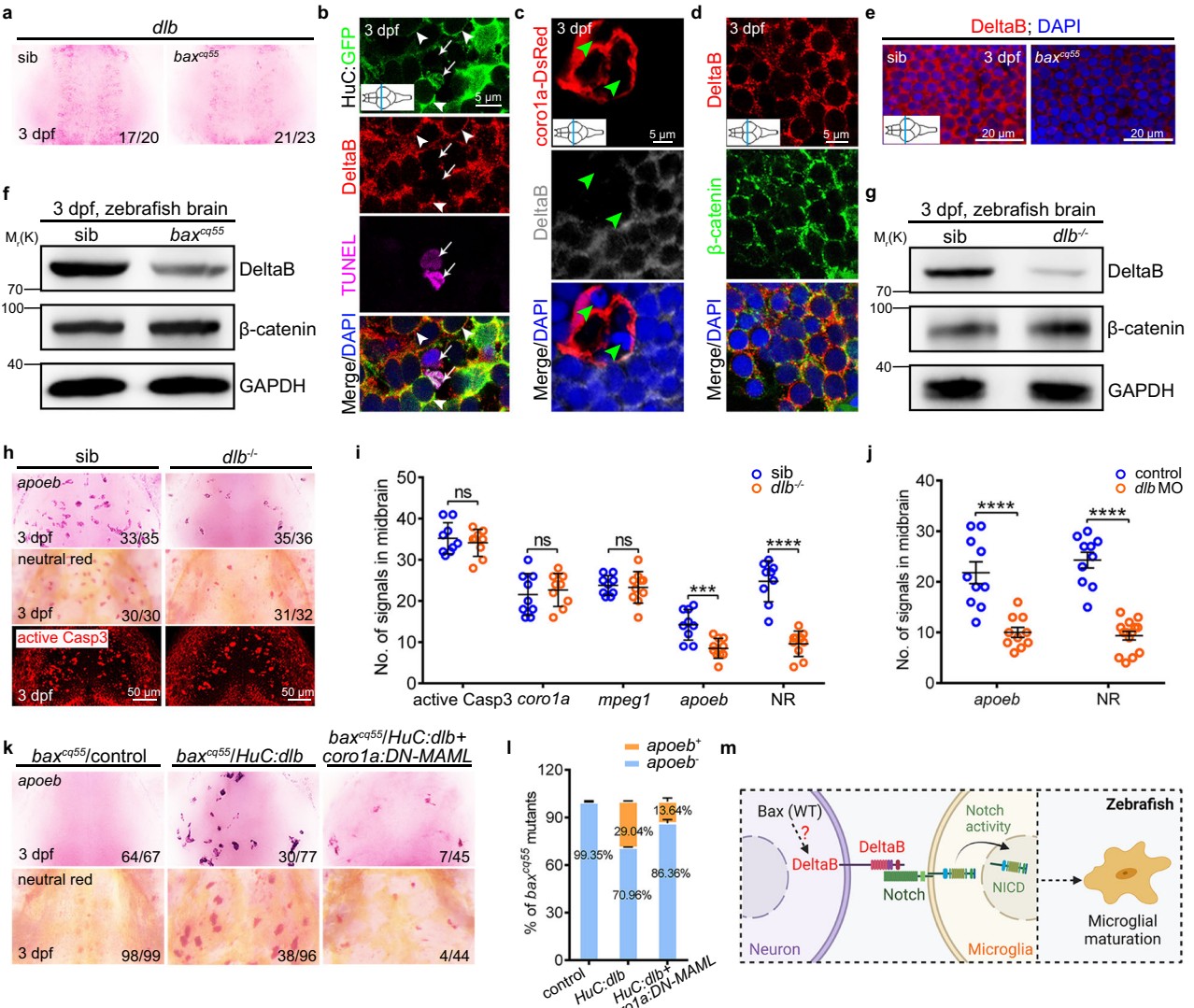

**Fig. 4 | Neuronal DeltaB serves as a key ligand triggering Notch activation in microglia. a** WISH of *dlb* in 3 dpf midbrains. **b–d** Immunofluorescence of HuC-GFP (green) with DeltaB (red) and TUNEL (magenta) (**b**) or coro1a-DsRed (red) with DeltaB (gray) (**c**) or DeltaB (red) with β-catenin (green) (**d**) in the transverse sections of 3 dpf midbrains. DeltaB mainly co-localized with the GFP⁺ signals in membrane (white arrowhead), rarely with DsRed⁺ (white arrow), neither TUNEL signals (white arrow), nor non-phagocytized cells (green arrowhead). **e** Immunofluorescence of DeltaB in the transverse sections of 3 dpf midbrains. **f, g** WB of DeltaB, β-catenin and GAPDH in *bax^cq55* (**f**) and *dlb^-/-* (**g**). The experiments were repeated three times and the results were similar. **h** Representative images of *apoeb* WISH, NR staining, and active Casp3 immunofluorescence in 3 dpf midbrains. **i, j** Quantification of active Casp3⁺, *coro1a⁺*, *mpeg1⁺*, *apoeb⁺*, and NR⁺ signals in siblings and *dlb^-/-* (**i**, active Casp3, *coro1a*: *n* = 9 per group; *mpeg1*, *apoeb*, NR: sib: *n* = 9, *dlb^-/-*: *n* = 10

per group) and *apoeb⁺* and NR⁺ signals in control and *dlb* MO group (**j**, *apoeb*: *n* = 10 per group; NR: control: *n* = 10, *dlb* MO: *n* = 14). Each dot denotes one fish. **k** Representative images of *apoeb* WISH and NR staining in the midbrains after supplying *HuC:dlb* or synergistic application of *HuC:dlb* and *coro1a:DN-MAML*. **l** The percentage of *bax^cq55* mutants with rescued *apoeb⁺* signals in (**k**). The number in each histogram indicates the average percentage. Data are pooled from three independent experiments. **m** A diagram of neuronal DeltaB inducing Notch activation in microglia, created with BioRender.com. WB, western blot. The inserted panels in (**b–e**) indicate the positions sectioned (blue line). Numbers in the right corners in (**a**, **h**, and **k**) indicate the counts of embryos with a typical appearance (first number) in the total examined fishes (last number). Error bars, mean ± SEM. ***P < 0.001; ****P < 0.0001; ns no significant, Unpaired two-tailed Student's *t* test. Source data are provided as a Source Data file.

which were quantified by the ΔF/F0 peak, declined notably in the HF GCaMP6s⁺ transients of *bax^cq55* mutants compared with those of the siblings (Fig. 5e). However, no apparent changes were detected in the population with a low frequency (LF, GCaMP6s⁺ transients occurring less than nine times) between the siblings and *bax^cq55* mutants (Fig. 5d, e, Supplementary Fig. 6c, and Supplementary Movie 4).

We next determined whether recovering the neuronal activities of Ca²⁺ signaling could restore the *dlb* expression and microglia phenotypes in the *bax^cq55* mutants. To this end, uncaging experiments were conducted by administrating cage IP3 or glutamate to the *bax^cq55* mutant midbrains (Supplementary Fig. 6d). The neuronal GCaMP6s⁺ dynamics, particularly those with HF, were recovered in the *bax^cq55*

mutants upon uncaging (Supplementary Fig. 6e, f, Supplementary Movie 5). Simultaneously, the transcript levels of *dlb* and the *apoeb⁺* and NR⁺ signals were effectively boosted in the uncaged *bax^cq55* mutant brains (Fig. 5f, g, Supplementary Fig. 6g). However, no evident intervention in the TUNEL⁺ signals was observed simultaneously (Fig. 5g). To perform a better estimation, nemadipine and nilvadipine, two calcium channel inhibitors, were utilized to disrupt the neuronal activities in WT embryos. Both chemicals perturbed the GCaMP6s⁺ dynamics and induced significant reductions in the *dlb⁺*, *apoeb⁺*, and NR⁺ signals (Fig. 5h, Supplementary Fig. 6h, i, Supplementary Movie 6). Notably, the c-D⁺ cells exhibited active and random motility in response to nemadipine, similar to the results observed in the *bax^cq55* mutants

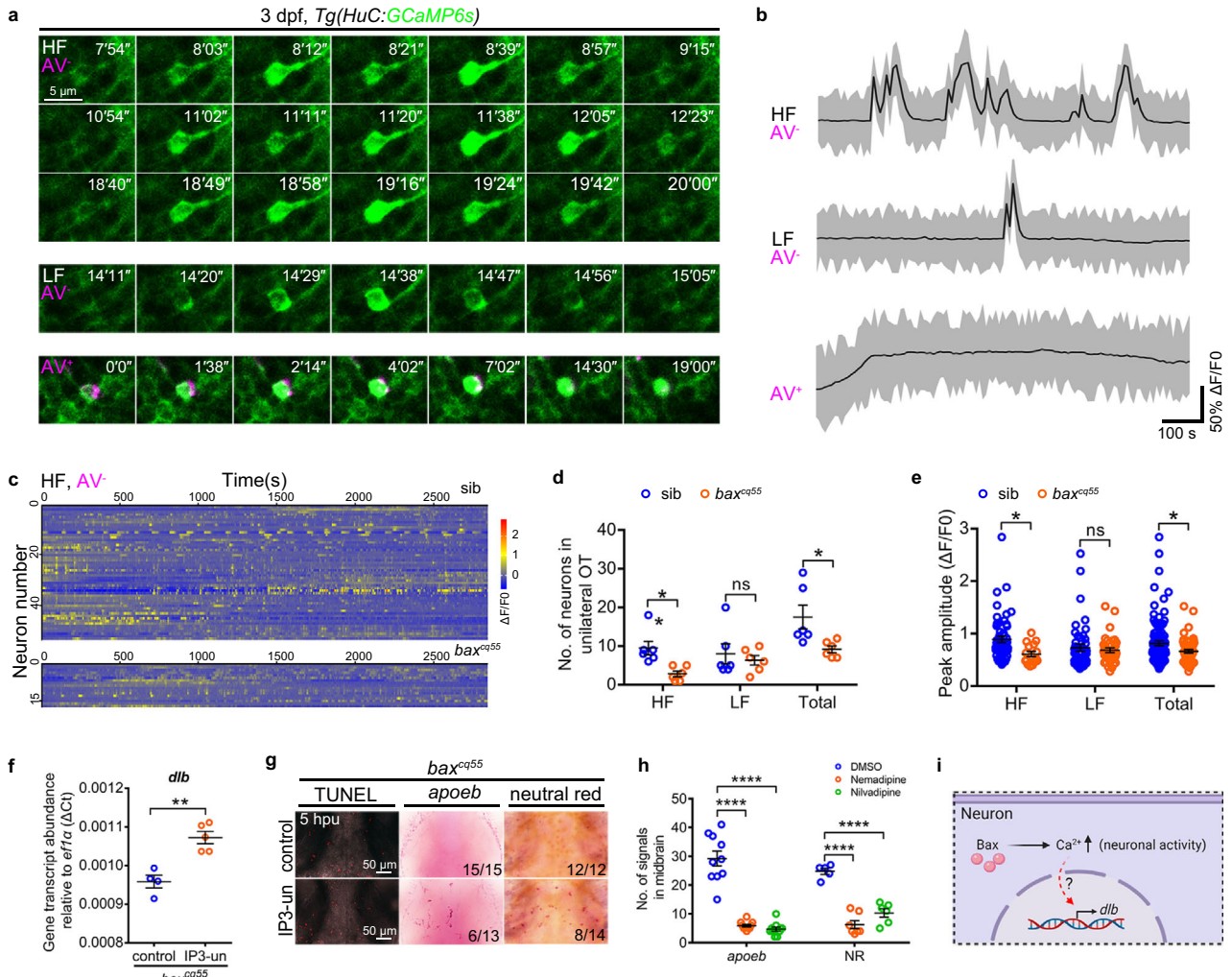

**Fig. 5 | Bax drives *dlb* expression via a special type of neuronal activities.**
**a** Confocal time-lapse imaging of GCaMP6s⁺ signals (green) in HF (top), LF (middle) and AV⁺ apoptotic neurons (bottom, magenta) in the midbrains. The number in the up-right corner indicated the time pointes. **b** Representative patterns of Ca²⁺ events in the HF, LF, and AV⁺ neurons during 1000 seconds of imaging. **c** GCaMP6s⁺ traces of HF neurons in the unilateral midbrains of 3 dpf siblings and *bax^{cq55}* mutants (*n* = 6 per group). **d** The number of HF, LF, and total active neurons in the unilateral midbrains of 3 dpf siblings and *bax^{cq55}* mutants. Each dot denotes one fish (*n* = 6 per group). **e** Quantification of the peak amplitude of calcium events (ΔF/F0) of HF, LF, and total calcium events in the unilateral midbrains of 3 dpf siblings and *bax^{cq55}* mutants (*n* = 6 fish in each group). Each dot denotes one calcium event. **f** qPCR result of the *dlb* transcriptional level at 5 hpu. The data are from three independent

experiments. Each dot represents an independent experiment. **g** Representative images of *apoeb* WISH, NR and TUNEL staining in the midbrains of *bax^{cq55}* mutant at 5 hpu. **h** Quantification of *apoeb*⁺ and NR⁺ signals after application of nemadipine and nivadipine. Each dot denotes one fish (*apoeb*: *n* = 10 per group; NR: DMSO: *n* = 5, nemadipine: *n* = 7, nilvadipine: *n* = 6). **i** Schematic presentation of Bax regulating *dlb* via neuronal activity, created with BioRender.com. HF high frequency, LF low frequency, AV Annexin V, IP3-un IP3 uncaging, hpu hours post-uncaging. Numbers in the right corners in (**g**) indicate the counts of embryos with a typical appearance (first number) in the total examined fishes (last number). Error bars, mean ± SEM. *P* < 0.05; **P* < 0.01; ****P* < 0.0001; ns no significant, Unpaired two-tailed Student's *t* test. Source data are provided as a Source Data file.

---

(Supplementary Fig. 6j, k, Supplementary Movie 7). These results indicate the instructive role of neuronal activity, which is regulated by BAX, in *dlb* expression and microglial development (Fig. 5i).

**BAX regulates *dlb* transcription through the CaMKII-CREB axis**
CaMKII, the Ca²⁺/calmodulin-dependent protein kinase type II, is an enzyme enriched in the central nervous system and was selected as a candidate target of Ca²⁺ signals in modulating *dlb*. The qualified p-CaMKII antibody was applied (Supplementary Fig. 7a). The autophosphorylated form of CaMKII was more obviously observed in H-G⁺ than in c-D⁺ cells (Fig. 6a). However, its activity diminished robustly in the *bax^{cq55}* mutants compared to that of their siblings (Fig. 6c, Supplementary Fig. 7b). KN-93 is a p-CaMKII inhibitor that specifically suppresses p-CaMKII activity but maintains the CaMKII global levels of the inactivated form (unphosphorylated)[40]. Treating WT embryos with KN-93 caused a significant reduction in *dlb* expression and the

simultaneous abortive generation of a-G⁺, *apoeb*⁺ and NR⁺ cells (Supplementary Fig. 7c–f). CREB is typically thought to be phosphorylated by activated p-CaMKII. The transcription of the factors in the CREB pathway was notably suppressed in the *bax^{cq55}* mutants (Fig. 3c). The widespread application of *creb1a*, rather than *creb1b* mRNA, or the CREB agonist Forskolin, in the *bax^{cq55}* mutants promoted the reappearance of *apoeb*⁺ and NR⁺ cells (Supplementary Fig. 7g). CREB is highly conserved[41], and the p-CREB antibody was observed to function effectively (Supplementary Fig. 7h). The immunofluorescence staining and western blot results revealed that p-CREB was highly expressed in the H-G⁺ cells (Fig. 6b), and that its activity was significantly erased in the *bax^{cq55}* mutants (Fig. 6c, Supplementary Fig. 7i). Disrupting the p-CREB activity by applying a specific inhibitor (PF-CBP1 HCl) or the newly designed *creb1a* MOs caused a significant depletion in both the *dlb* expression and c-D⁺ pools, as well as the a-G⁺c-D⁺ and NR⁺ populations (Fig. 6d, e, Supplementary Fig. 7j–o). Furthermore, the

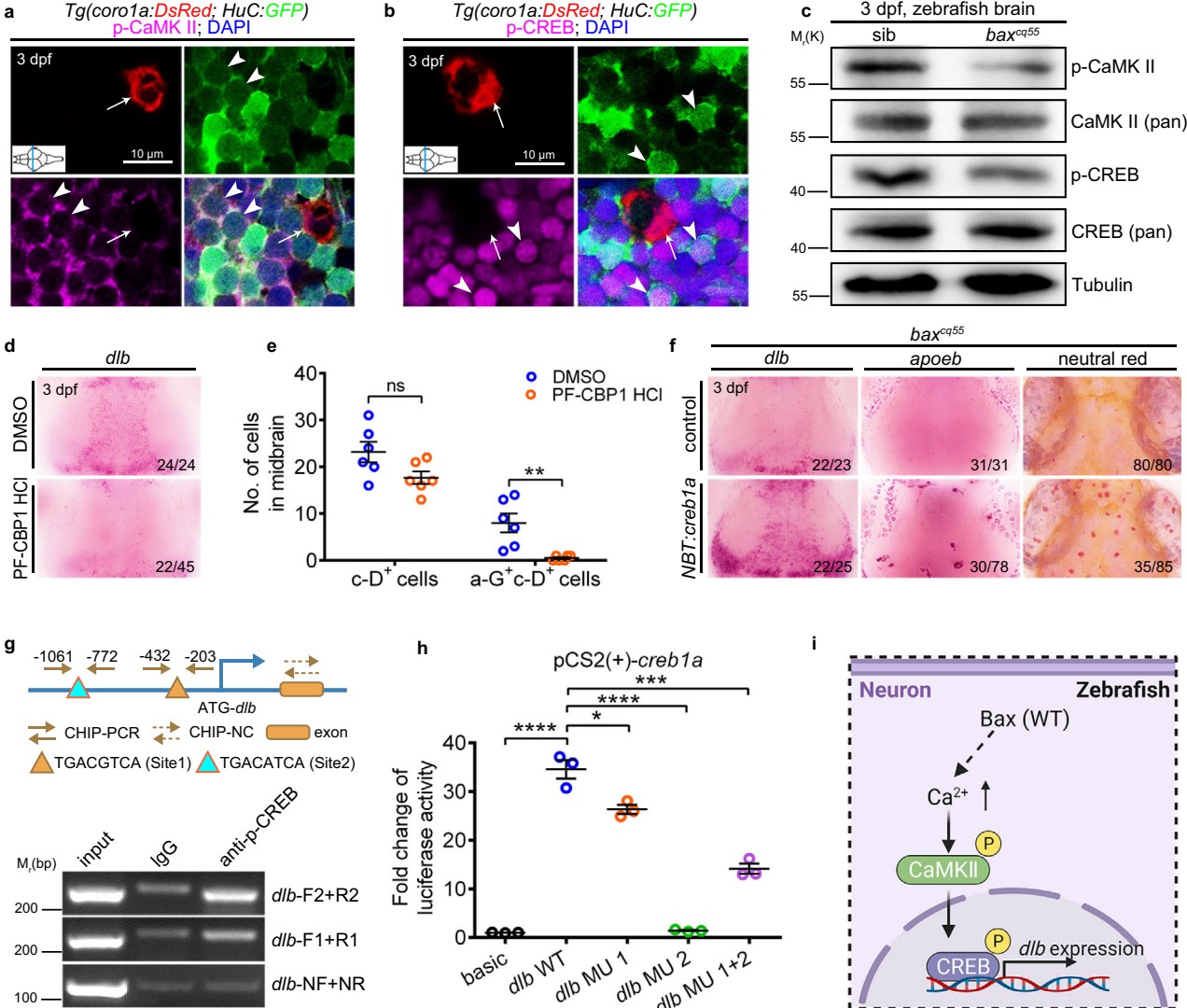

**Fig. 6 | The CaMKII-CREB pathway regulates *dlb* expression and microglial maturation. a** Immunofluorescence of p-CaMKII (magenta), H-G (green) and c-D (red) in the transverse sections of 3 dpf WT midbrain. p-CaMKII co-stained with GFP⁺ (white arrowhead) but not DsRed⁺ (white arrow) signals. **b** Immunofluorescence of p-CREB (magenta), H-G (green) and c-D (red) in the transverse sections of 3 dpf WT midbrain. p-CREB merged with GFP⁺ (white arrowhead) but not DsRed⁺ (white arrow) signals. **c** WB of p-CaMKII, CaMKII (pan), p-CREB, CREB (pan), and Tubulin in the brain cells from 3 dpf siblings and *bax^{cq55}* mutants. The experiments were repeated three times and the results were similar. **d** WISH of *dlb* in 3 dpf midbrains after treated with DMSO or PF-CBP1 HCl. **e** The number of c-D⁺ and a-G⁺c-D⁺ signals in 3 dpf midbrains after DMSO and PF-CBP1 HCl treatment. Each dot denotes one fish (*n* = 6 in each group). **f** Representative images of *dlb* and *apoeb* WISH, and NR staining in the *bax^{cq55}* midbrains after supplying *NBT:creb1a*. **g** Top panel: the

predicted schematic structure of *dlb* promoter with two potential CREB binding sites. Bottom panel: the gel results of ChIP-PCR. p-CREB binding sites 1 and 2 were amplified by the *dlb*-F1/R1 and *dlb*-F2/R2 primers, respectively. *dlb*-NF/NR primers were used as the negative control. **h** Quantification of fold alterations in the luciferase activities after transfected with different vectors. Data were pooled from three independent experiments. Each dot represents an independent experiment. **i** A diagram of the CaMKII-CREB axis in controlling *dlb* expression, created with BioRender.com. MU, mutation. Numbers in the right corners in (**d**) and (**f**) indicate the counts of embryos with a typical appearance (first number) in the total examined fishes (last number). Error bars, mean ± SEM. *P < 0.05; **P < 0.01; ***P < 0.001; ****P < 0.0001; ns no significant, Unpaired two-tailed Student's *t* test. Source data are provided as a Source Data file.

introduction of *NBT:creb1a* in the *bax^{cq55}* mutant NBT⁺ cells significantly elevated the *dlb* transcript levels and effectively rescued the *apoeb*⁺ and NR⁺ cells in approximately 41% of the embryos (Fig. 6f). Two potential CREB binding sites in the *dlb* promoter (Site 1: TGACGTCA; Site 2: TGACATCA; Fig. 6g) were predicted using bioinformatics. Concordantly, the DNA fragments containing the suggested binding elements were immunoprecipitated using an anti-p-CREB antibody (Fig. 6g). The overexpression of a full-length *creb1a* coding sequence (CDS) in HEK293T cells significantly elevated the luciferase activity of the *dlb* promoter harboring two p-CREB binding sites (Fig. 6h). Conversely, the luciferase activity was reduced when the p-CREB binding sites, especially that of site 2, were mutated (Fig. 6h), supporting that

*dlb* was directly targeted by activated CREB. Thus, BAX regulates *dlb* transcription via the CaMKII-CREB axis in embryonic neurons (Fig. 6i).

## *Bax^{tm1Sjk}* mutant mice exhibit compromised microglial maturation and Notch signaling in embryonic stages

The effective mitigation of microglia deficiency via the supply of mouse, or even human BAX, to *bax^{cq55}* mutant zebrafish embryos suggested a conserved regulatory axis (Supplementary Fig. 2f). Correspondingly, the mouse BAX level was higher in E14.5 embryonic SOX2⁺ than in F4/80⁺ cells in the mesencephalic region (Fig. 7a, Supplementary Fig. 8a). We analyzed the microglia density, morphology, and identity in the brain tissues of E14.5 *Bax^{tm1Sjk}* mice, which presented

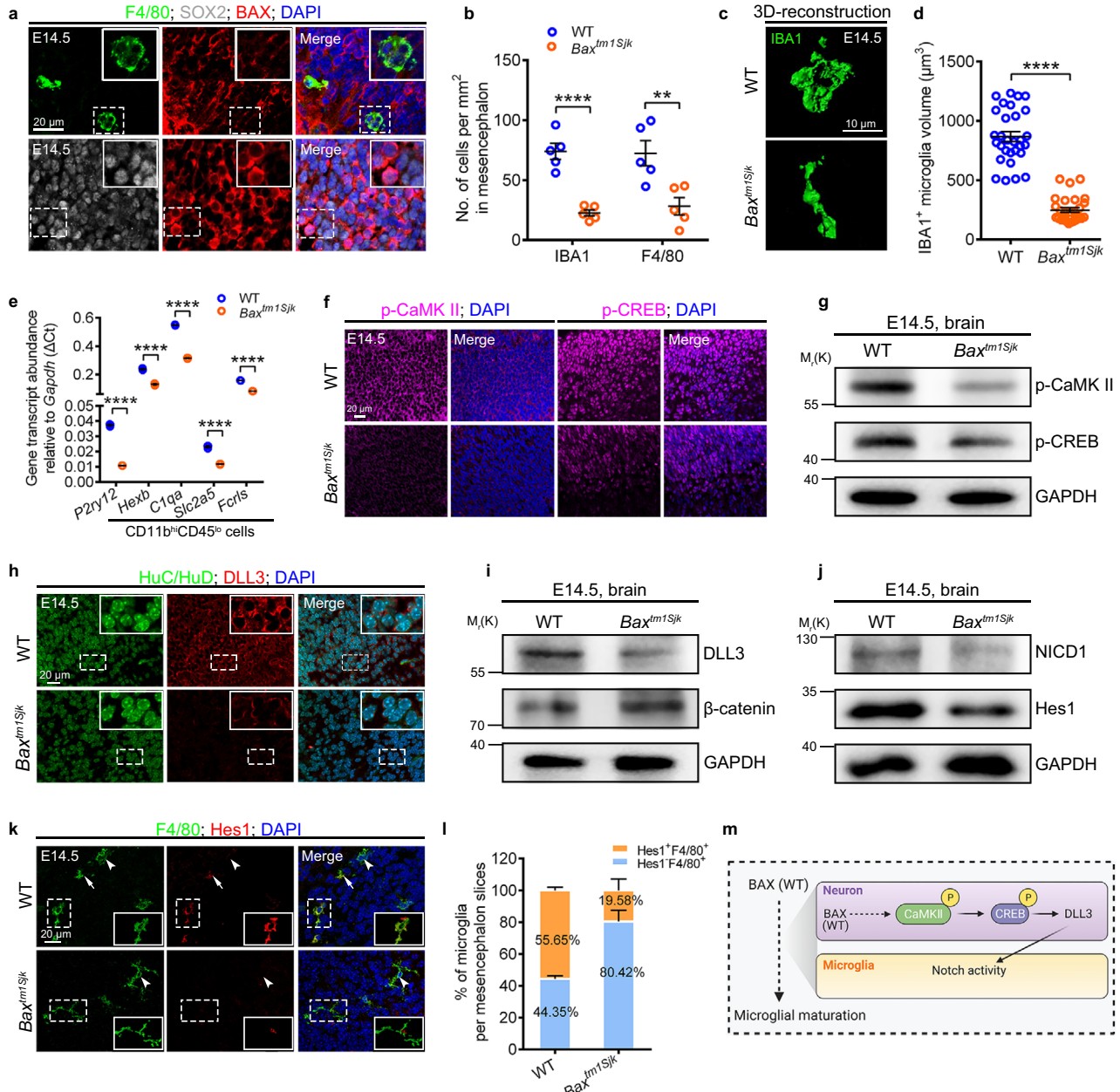

**Fig. 7 | *Bax^{tm1Sjk}* mutant mice exhibit compromised microglial maturation and Notch signaling. a** Immunofluorescence of BAX (red) with SOX2 (gray) and F4/80 (green) in the sagittal sections of E14.5 mice brains. **b** Quantification of IBA1+ and F4/80+ cell density in E14.5 WT and *Bax^{tm1Sjk}* mice midbrains. Each dot denotes one mouse (*n* = 5 mice in each group). Three sections per mouse were quantified. **c, d** 3D reconstructions of confocal z-stacks (**c**) and measured volume (**d**) of IBA1+ cells in E14.5 WT and *Bax^{tm1Sjk}* mice mesencephalon (*n* = 3 mice in each group). Each dot in (**d**) denotes one cell. Ten cells per mouse were quantified. **e** Transcriptional levels of *P2ry12*, *Hexb*, *C1qa*, *Slc2a5*, and *Fcrls* in CD11b^{hi}CD45^{lo} cells from E14.5 WT and *Bax^{tm1Sjk}* mice mesencephalon. The data are from three independent experiments. Each dot represents an independent experiment. **f** Immunofluorescence of p-CaMKII (left) and p-CREB (right) in brain sagittal sections of E14.5 WT and *Bax^{tm1Sjk}* mice. **g** WB of p-CaMKII, p-CREB and GAPDH in the brain cells from E14.5 WT and *Bax^{tm1Sjk}* mice. The experiments were triply repeated. The results were similar.

**h** Immunofluorescence of HuC/HuD (green) and DLL3 (red) in the brain sagittal sections of E14.5 WT and *Bax^{tm1Sjk}* mice. **i, j** WB of DLL3 and β-catenin (**i**), NICD1 and Hes1 (**j**) in the brain cells from E14.5 WT and *Bax^{tm1Sjk}* mice. The experiments were triply repeated. The results were similar. **k** Immunofluorescence of F4/80 (green) and Hes1 (red) in the brain sagittal sections of E14.5 WT and *Bax^{tm1Sjk}* mice. White arrows indicated the co-stained signals. White arrowheads presented the F4/80+ signals only. **l** The percentage of F4/80+ microglia with or without Hes1+ signals in (**k**) (*n* = 4 mice in each group). Three sections per mouse were quantified. The number in each histogram indicates the average percentage. **m** A diagram of the Bax-Notch axis in regulating microglial maturation, created with BioRender.com. The small images in the right corners of (**a**, **h**, and **k**) are the enlargement regions (white dotted box). Error bars, mean ± SEM. **\*\**P* < 0.01; \*\*\*\**P* < 0.0001, Unpaired two-tailed Student's *t* test. Source data are provided as a Source Data file.

a prominent absence of active Casp3+ signals (Supplementary Fig. 8b, c). Like those in *bax^{cq55}* mutant zebrafish, the density and population of either F4/80+ and IBA1+ or CD11b^{hi}CD45^{lo} cells were significantly reduced in the E14.5 *Bax^{tm1Sjk}* mesencephalon (Fig. 7b, Supplementary Fig. 8d–g). The limited IBA1+ cells in *Bax^{tm1Sjk}* embryos presented a long

thin appearance, and their sizes were much smaller than those of the amoeboid WT counterparts (Fig. 7c, d). Furthermore, the examination of the microglial homeostatic signature genes (MHSGs) (*P2ry12*, *Hexb*, *C1qa*, *Slc2a5*, and *Fcrls*)[16] revealed a significant decline in their transcript levels in the remaining CD11b^{hi}CD45^{lo} cells from *Bax^{tm1Sjk}* mice

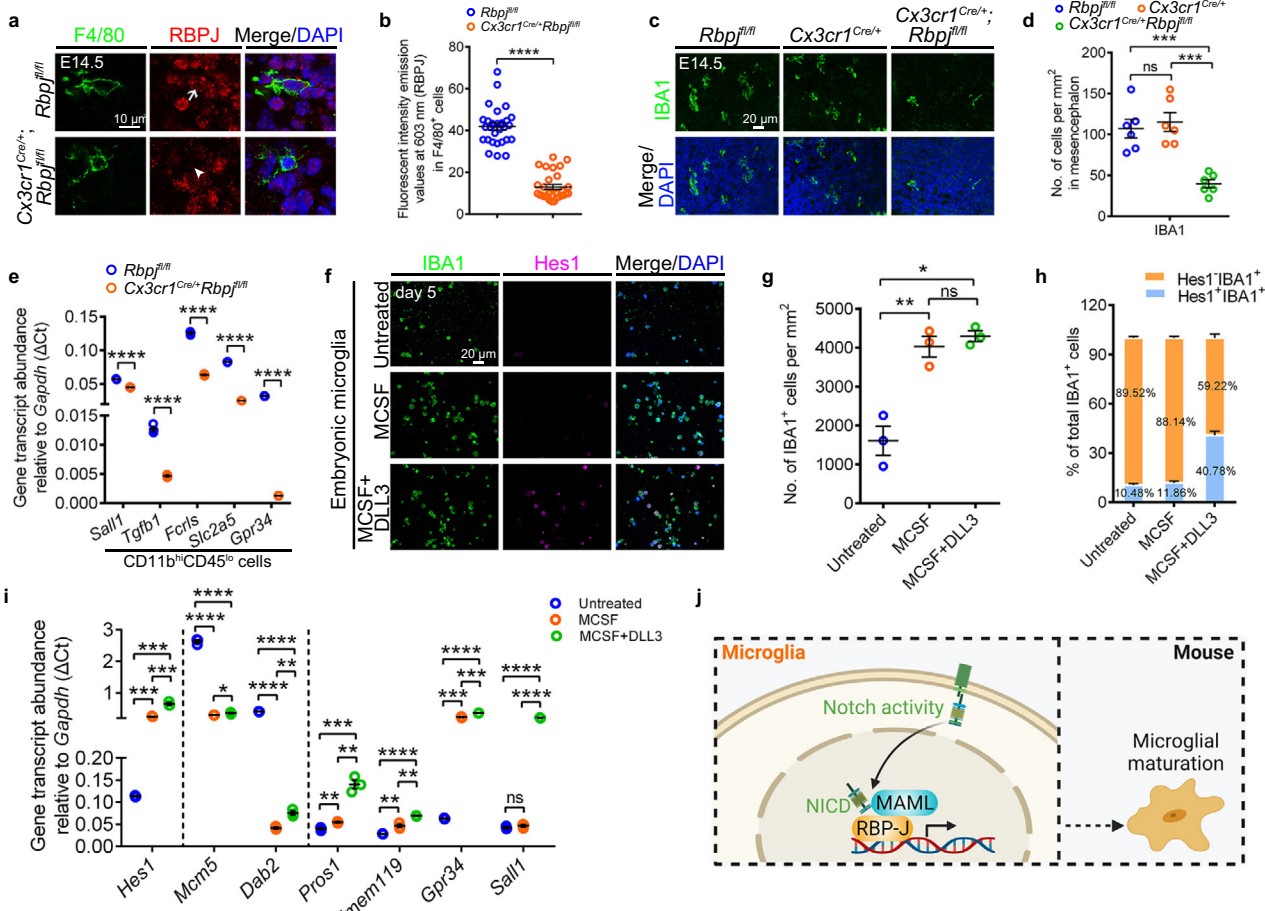

**Fig. 8 | Notch activity promotes microglial development in vivo and in vitro.**
**a**, **b** Immunofluorescence (**a**) and fluorescent intensity emission values (**b**) of RBPJ (red) in F4/80⁺ (green) cells in the sagittal sections of E14.5 *Rbpj^fl/fl* and *Cx3cr1^cre/+ Rbpj^fl/fl* mice (n = 3 mice in each group). Each dot in (**b**) denotes one cell. Ten cells per mouse were quantified. The white arrow indicates the co-localized signals of RBPJ and F4/80, whereas white arrowhead point to the limited signals of RBPJ.
**c**, **d** Immunofluorescence (**c**) and quantified cell density (**d**) of IBA1⁺ cells in the sagittal sections of E14.5 *Rbpj^fl/fl*, *Cx3cr1^cre/+* and *Cx3cr1^cre/+Rbpj^fl/fl* mice (n = 6 mice in each group). Each dot denotes one mouse. Three slices per mouse were quantified.
**e** Transcriptional levels of *Sall1*, *Tgfb1*, *Fcrls*, *Slc2a5*, and *Gpr34* in CD11b^hiCD45^lo cells from E14.5 *Rbpj^fl/fl* and *Cx3cr1^cre/+Rbpj^fl/fl* mice mesencephalon. The data are from three independent experiments. Each dot represents an independent experiment.
**f** Immunofluorescence of IBA1 and Hes1 in the embryonic microglia in vitro

furnished with MCSF or additional DLL3 for 5 days. **g** Quantification of IBA1⁺ cell density in (**f**). Data were pooled from three independent experiments. Cultured IBA1⁺ cells in each experiment were from twenty E12.5 mice mesencephalon. Each dot represents individual experiment. **h** The percentage of Hes1⁻IBA1⁺ or Hes1⁺IBA1⁺ cells after cultured 5 days in (**f**). The number in each histogram indicates the average percentage. Data are pooled from three independent experiments.
**i** Transcriptional levels of *Hes1*, *Mcm5*, *Dab2*, *Pros1*, *Tmem119*, *Gpr34*, and *Sall1* in embryonic microglia after cultured for 5 days. The data were from three independent experiments. Each dot represents an independent experiment.
**j** Schematic diagram of Notch activation in mouse microglial differentiation, created with BioRender.com. Error bars, mean ± SEM. *P < 0.05; **P < 0.01; ***P < 0.001; ****P < 0.0001; ns no significant, Unpaired two-tailed Student's *t* test. Source data are provided as a Source Data file.

(Fig. 7e). p-CaMKII, p-CREB, and DLL3, which were more easily discerned in HuC/D⁺ neurons than in F4/80⁺ microglia, were significantly decreased in the *Bax^tm1Sjk* brains compared with their amount in the controls (Fig. 7f–i, Supplementary Fig. 8h, i). Concurrently, the levels of NICD1 subsided significantly in *Bax^tm1Sjk* mice (Fig. 7j). Hes1, a Notch target gene, was detected in approximately 55.65% of the F4/80⁺ cells in the E14.5 WT mesencephalon (Fig. 7k, l). However, it was sparsely found in 19.58% of the remaining counterparts in *Bax^tm1Sjk* (Fig. 7k, l). The transcripts level of *Hes1* was significantly reduced in the CD11b^hiCD45^lo cells in the E14.5 *Bax^tm1Sjk* brains (Supplementary Fig. 8j). These findings imply the conserved regulatory mechanisms of the BAX-Notch axis in the developmental programming of microglia (Fig. 7m).

## Microglia-specific Notch deficiency impairs embryonic microglial maturation

To validate the role of Notch activity in mouse microglial development, we reviewed the bioinformatics data regarding the

CD45^lowCx3cr1^highCD11b^high cells from E14.5 mice[42]. The unsupervised clustering of these cells revealed 13 unique clusters (Supplementary Fig. 8k). In the twelve subpopulations of microglia signatures, the transcript levels of the Notch signaling factors were impressively higher than those of their border-associated macrophages partners (Supplementary Fig. 8l, m). We disrupted the Notch activity in mouse microglia by generating *Cx3cr1^Cre/+Rbpj^fl/fl* mice via crossing conditional alleles of *Rbpj* with a microglia-specific Cre-recombinase line. The activities of RBPJ and Hes1 decreased significantly in the F4/80⁺ and IBA1⁺ cells from *Cx3cr1^Cre/+Rbpj^fl/fl* mice compared with those of the controls at E14.5 (Fig. 8a, b, Supplementary Fig. 9a–c). No visible changes in individual development were observed in *Cx3cr1^Cre/+Rbpj^fl/fl* embryonic mice. We excluded the influence of Cx3cr1-Cre[43] in embryonic microglial development, according to the paralleled populations of IBA1⁺ and F4/80⁺ cells, the percentage of CD11b^hiCD45^lo microglia, and their gene expression in the mesencephalic regions among *Rbpj^fl/fl*, *Cx3cr1^Cre/+*, and WT mice (Fig. 8c, d, Supplementary Fig. 9d–h). However, the populations of IBA1⁺ and F4/80⁺ cells were

notably diminished in the mesencephalic regions of E14.5 $Cx3cr1^{Cre/+}Rbpj^{fl/fl}$ mice compared to those of either their $Rbpj^{fl/fl}$ or $Cx3cr1^{Cre/+}$ siblings (Fig. 8c, d, Supplementary Fig. 9d, e). Consistently, flow cytometry results revealed a lower portion of the CD11b$^{hi}$CD45$^{lo}$ pools in the brains of $Cx3cr1^{Cre/+}Rbpj^{fl/fl}$ mice than in the $Rbpj^{fl/fl}$ controls (Supplementary Fig. 9i, j). The residual IBA1$^+$ cells in E14.5 $Cx3cr1^{Cre/+}Rbpj^{fl/fl}$ mice presented a typical amoeboid appearance; however, they were smaller than their $Rbpj^{fl/fl}$ counterparts (Supplementary Fig. 9k, l). Of note, the MHSGs (Sall1, Tgfb1, Fcrls, Slc2a5, and Gpr34) transcript levels were significantly lower in the CD11b$^{hi}$CD45$^{lo}$ cells of $Cx3cr1^{Cre/+}Rbpj^{fl/fl}$ mice than in the $Rbpj^{fl/fl}$ controls (Fig. 8e).

### Notch activation contributes to the microglia signature in vitro

To further corroborate the prominence of Notch activity, we sorted the CD11b$^+$ cells from E12.5 embryonic mice cerebral cortexes (largely immature microglia at embryonic stages) and cultured them in a microglia nutrition medium containing MCSF or MCSF plus DLL3 (Supplementary Fig. 10a, b). The addition of MCSF alone led to a noticeable increase in the survival and proliferation of IBA1$^+$ cells after two days (Supplementary Fig. 10c–e). After five days, these IBA1$^+$ cells cultured with MCSF presented lower expression levels of early microglia genes (Mcm5 and Dab2) but elevated levels of Hes1 and MHSGs (Pros1 Tmem119, and Gpr34) (Fig. 8f–i). Notably, the additional supplementation with DLL3 led to a drastic expansion of the Hes1$^+$ signals in the similarly enlarged IBA1$^+$ population compared with that of those cultured with MCSF alone. However, these cells presented a much more significant elevation of the MHSGs (Pros1, Tmem119, Gpr34, and Sall1) levels than their counterparts cultured in a medium supplemented with MCSF alone (Fig. 8i). These in vivo and in vitro results together substantiate the important role of Notch signaling in triggering embryonic microglia differentiation in mice (Fig. 8j).

## Discussion

The optical transparency of zebrafish embryos provides advantages for the live imaging and precise delineation of nascent microglial maturation. The initial long thin PM actively moved within the brain. They attained the microglia signatures to differentiate into AM and engulfed apoptotic neurons. The body enlargement and mobility reduction resulting from phagocytosis plausibly facilitate the intimate and effective conversation between microglial cells and the brain milieu to gain their resident identity during maturation (Supplementary Fig. 10f). A similar phenomenon was noticed in mice. Transcriptomic and functional studies revealed that the mouse microglia underwent distinct phases of differentiation that relied on the expression of milieu cytokines, including but not restricted to Cxcl14, Csf1, and Il-34[15,19,44], which are derived from neurons. These observations support the sheared cellular features of environmental components in promoting microglial development in zebrafish and higher vertebrates.

The exposure of microglia to astrocyte/neuron co-cultures conditioned medium (AN-CM) promotes their maturation with an increased ramification and a significant upregulation of MHSGs. However, microglia signatures subside rapidly in culture, evidencing that microglial maturation requires additional instructions. Notch activity was detected in the embryonic microglia in both zebrafish and mice. The specific intervention of Notch activity in zebrafish and mice significantly compromised microglial maturation in the embryonic stages, substantiating the sufficient roles of Notch signaling in early microglia differentiation. The Notch pathway functioned in the cell–cell contact scenario. The enlarged size of microglia and the enhanced neuronal density during embryonic development contributed to the increase in the number of neurons connecting with microglia, which agreed with the role of Notch in the microglial maturation process. In the macrophage and/or microglial development, Notch signaling fosters macrophage maturation when

challenged with pro-inflammatory stimuli[45]. Notch signaling is also involved in microglia activation in neuro-inflammatory diseases[23]. Our results extended the understanding of the indispensable roles of Notch in nascent microglial maturation. Concordantly, the in vitro culture of the isolated embryonic mice microglia with DLL3 and MCSF aggregated the Notch activity, which could increase the expression of several MHSGs. However, we did not estimate the influence of TGF-β, a verified factor in harnessing adult microglia signatures and functions[20], on this condition. Future studies should primarily focus on the comparison and cooperation of TGF-β with Notch pathways in microglial development, which might be valuable in optimizing microglia culture.

The dlb level decreased the most among those of the Notch ligands in the $bax^{cq55}$ mutant neurons. Further investigation indicated that dlb was expressed in the living neuron membranes and was transcriptionally controlled by p-CREB, which is regulated by the neuronal activity-dependent calcium-CaMKII axis[46]. The neuronal activity attracts microglia and regulates their behaviors. When the voltage-gated sodium channel blocker tetrodotoxin (TTX) is injected into the zebrafish OT, it suppresses neural activity and reduces the contact of the microglia bulbous with neurons[47]. The current imaging results of the neuronal activity, which were reflected by the presentation of calcium events, appeared at 2.5–3 dpf in zebrafish, suggesting the appearance of functional neurons at this stage. This process was consistent with microglial development. Several types of neuronal activities showing different calcium signal frequencies and aptitudes were primarily classified. The HF subtype of neuronal activity was altered in the $bax^{cq55}$ mutant brains, suggesting that a specific neuronal subtype or condition plays a functional role in early microglial development. However, whether the alterations in neuronal activity affect the behavior of $bax^{cq55}$ mutant fish and their underlying cellular and molecular basis requires further investigation. The membrane localization of dlb facilitated the transactivation of Notch signaling through effective contact with microglia for their identity acquisition. A compromised dlb function in the mutant or the MO assay led to a significant reduction, but not total loss of, apoeb$^+$ microglia compared to that observed in the $mib^{taS2b}$ embryos. This difference was probably ascribed to the redundant roles of other ligands in the $bax^{cq55}$ mutant.

A pronounced reduction in the density and expression of disease-associated microglia (DAM)-related genes was observed in the retinal microglia of Bax mutant mice, implying a major driver of the neuronal apoptosis in the disease-related profile in microglia[27]. However, our study extended the involvement of BAX in embryonic microglial maturation. BAX regulates neuronal apoptosis through its classical functions[27]. Dead cells are engulfed and cleared by microglia, which advances neural development by the refinement of neuronal components, networks, and even environments. However, microglia also benefit from this process. Their movement is reduced, and their bodies become enlarged from their active premature conditions, which advances their conditions to interact with living neurons and accept the instructive signals (Notch ligands). Moreover, BAX equips living neuronal cells with Notch ligands by regulating the calcium-CaMKII-CREB axis. The dual nature of BAX aligns well with embryonic microglial development and neurogenesis. Recognizing the molecular basis of the neuronal milieu in regulating one early step of microglial maturation will facilitate an extensive understanding of microglial development and provide clues regarding the pathogenesis of neurodegenerative diseases. This conclusion also suggests the application of microglia to perfect the existing therapeutic strategies.

## Methods

### Zebrafish and Mice

The wild-type (WT) AB and WIK strains were used in this study. The $mib^{taS2b}$ mutant[48], Tg(coro1a:DsRed)[31], Tg(coro1a:Kaede)[49], Tg(mfap4:GFP)[50], Tg(apoeb:GFP)[30], Tg(ubiq:secAnnexinV-mVenus)[51];

*Tg(Tp1bglob:eGFP)*[52], *Tg(Tp1bglob:hmgb1-mCherry)*[52], *Tg(hsp70:DN-MAML-GFP)*[53], *Tg(HuC:GCaMP6s)*[54] were employed. Zebrafish were bred and maintained under standard laboratory conditions according to Institutional Animal Care and Use Committee protocols. Embryos were raised in egg water at 28.5 °C and treated with 0.003% 1-phenyl-2-thiourea (PTU, Sigma-Aldrich, P7629) to inhibit pigment formation. Mice (C57BL/6J background) were maintained in a specific pathogen-free (SPF) condition in accordance with the guidelines of Institutional Animal Care and Use Committee protocols. All mice used were socially housed under a 12 h light/dark cycle with 40-60% relative humidity at 23 °C. The following mice were obtained from the Jackson laboratory: *Bax*[+/tm1Sjk] mice (B6.129×1-*Bax*[tm1Sjk]/J, 002994), *Rbpj*[fl/fl] (C57BL/6J-*Rbpj*[em2Lutzy]/J, 034200), *Cx3cr1-Cre* (B6J.B6N(Cg)-*Cx3cr1*[tm1.1(cre)Jung]/J, 025524). Mice carrying *Cx3cr1-Cre* transgene were crossed with *Rbpj*[fl/fl] mice to obtain *Cx3cr1-Cre, Rbpj*[fl/fl], and *Cx3cr1*[cre/+]*Rbpj*[fl/fl] mice. Mice were genotyped using polymerase chain reaction (PCR). And the toe DNA was used as a template. The genotyping primers' information is listed in Supplementary Table 1. All animal experiments were approved by the Institutional Review Board of Southwest University (Chongqing, China).

## Forward genetic screening and genetic mapping

The ENU mutagenesis screening and positional cloning were performed as described previously[35]. Briefly, adult WT zebrafish males of AB background were mutagenized with N-ethyl-N-nitrosourea (ENU) and then outcrossed to generate F1 and F2 families to perform the screening. A total of 203 crossing pairs from 42 F2 families were examined by *apoeb* WISH. Among the identified candidate mutant pairs with notable *apoeb* phenotypes at 3 dpf, the 55th pair was a recessive mutant allele showing loss of *apoeb* signals. Initial complementary assay indicated that this candidate mutant was not caused by *csf1ra*, *pu.1*, and *irf8* mutation and was nominated as cq55 (Chongqing (cq) NO. *55*) mutant. The *cq55* mutants were mated with WT female of WIK background to perform the positional cloning. We first mapped the mutation to linkage group (LG) 3 by bulked segregation analysis, which indicated a close correlation of the *cq55* mutant pools with the sequence-length polymorphism (SSLP) and single nucleotide polymorphism (SNP) markers in LG3. Based on the recombination events in *cq55* mutant embryos, high-resolution mapping was conducted by using additional polymorphic SSLPs and SNPs to narrow *cq55* mutation to a restricted region. Sequencing was eventually conducted to identify a T to A mutation in exon 4 of *baxa* gene (Ensemble: ENSDARG00000020623), which causes the alteration of Ile (I) to Asn (N) in the conserved BH1 domain in *cq55* mutants.

## Generation and genotyping of *bax* and *dlb* mutants

The *bax*[Δ1/Δ1], *bax*[Δ58/Δ58] and *dlb*[-/-] mutant alleles were generated by the CRISPR/Cas9 system[55]. The guide RNAs targeting the "gRNA1: 5′-GGACATCGTCCAGCTTATGA-3′ and gRNA2: 5′-GGATCAGGGAACAGGGTGGA-3′" in *bax* exon5 and 5′-GGAGAGTGCAAGTGTCGTCT-3′" in *dlb* exon5 were in vitro synthesized respectively using the T7 transcription kit (Thermo Fisher, AM1334). 1 nL of the mixed solution containing Cas9 mRNA (300 ng/μl) and gRNA (100 ng/μl) was injected into the fertilized embryos at one-cell stage. Two *bax* mutant (*bax*[Δ1/Δ1] and *bax*[Δ58/Δ58]) and one *dlb* mutant (2 bp deletion with a 10 bp insertion) alleles were identified by high-resolution melt analysis of PCR and DNA sequencing using the designed primers (seen in Supplementary Table 1).

## Whole-mount in situ hybridization (WISH), fluorescence in situ hybridization (FISH), neutral red (NR), acridine orange (AO), TUNEL, and immunofluorescence staining

The RNA probes were synthesized in vitro by using the DIG RNA Labelling kit SP6/T7 (Roche, 11175025910). WISH and FISH were performed following the standard protocols[49]. For the chemical staining assays, zebrafish embryos were incubated with NR solution[10] (Santa Cruz, sc-281691, 1:400) for 4 h and AO solution[12] (Sangon Biotech, A642007, 1:100) for 1 h. They were then washed with fresh E3 buffer before imaging. To perform the immunofluorescence staining, vibratome section of zebrafish embryos was applied using a Leica VT1000S machine according to a previous study[56]. Briefly, 3 dpf embryos were fixed by 4% paraformaldehyde (PFA) at room temperature (RT) for 1 h. They were dehydrated in 30% (M/V) sucrose at RT for 30 min and embedded in 4% low-melting-point agarose gel (LMA) for the followed section. The samples were sectioned to about 100 μm in thickness. Sagittal frozen section of E14.5 mouse brain was done as that described previously[57] by using a Leica CM1950 machine. The thickness of one slice was 10 μm. For immunofluorescence staining, the tissue pieces were rinsed in PBDT (PBS with 1% BSA, 1% DMSO, 0.5% TrintonX-100) and blocked with buffer (2% FBS and 0.1% Tween20 in PBS) at RT for 1 h. Primary antibodies were utilized at 4 °C overnight: goat anti-GFP (Abcam, ab6658, 1:400), mouse anti-DsRed (Santa Cruz, sc-101526,1:400), rabbit anti-DsRed (Takara Bio Clontech, 632496, 1:400), rabbit anti-Lcp1 (GTX124420, GeneTex, 1:400), mouse anti-zBax (This paper, 1:400), mouse anti-Bax (B-9) (Santa Cruz, sc-7480, 1:200), rabbit anti-activated Caspase-3 (BD Biosciences, 559565, 1:400), rabbit anti-SOX2 (Abcam, ab94959, 1:200), rabbit anti-CaMKII (pan) (Cell Signalling technology, 4436, 1:400), rabbit anti-phospho-CaMKII (Thr286) (Cell Signalling technology, 12716, 1:400), rabbit anti-CREB (Abcam, ab32515,1:400), rabbit anti-CREB (phospho S133) (Abcam, ab32096,1:400), rabbit anti-DLL3 (Abcam, ab103102, 1:200), rabbit β-catenin (Merck Millipore, 06-734, 1:400), Rat anti-F4/80 (Abcam, ab6640, 1:100), goat anti-IBA1 (Abcam, ab5076, 1:200), mouse anti-Hes1 (Abcam, ab119776, 1:400), rabbit anti-activated Notch1 (Abcam, ab8925, 1:400), rabbit anti-Rbpj (BOSTER, A00767-1, 1:400), mouse anti-FLAG (Sigma-Aldrich, F1804, 1:400). Afterwards, the samples were washed with PBDT and then stained with AF488/555/647-conjugated secondary antibodies including 488 Goat Anti-Rat (Abcam, ab150165), 488 Donkey Anti-Goat (Thermo Fisher Scientific, A-11055), 555 Donkey Anti-Rabbit (Thermo Fisher Scientific, A-31572), 647 Donkey Anti-Mouse (Thermo Fisher Scientific, A-31571) or TUNEL by its specific kit (Roche, 12156792910) at 37 °C for 3 h. The WISH pictures were captured by a Carl Zeiss Discovery.V20 microscope. FISH and immunofluorescence staining images were taken by Zeiss LSM700 and LSM880 confocal microscopes.

## Uncaging of IP3 and Glutamate

1 mM NPE-caged-IP3 (Invitrogen, I23580) or 5 mM MNI-caged-L-glutamate (Tocris Bioscience, 1490) was mixed with pluronicF127 (final concentration 0.2%) (Invitrogen, p-6866) and Alexa Fluor 647 dextran (Invitrogen, D22914, 1:1000). The mixture was injected into the midbrain of anesthetized 3dpf *bax*[cq55]/*Tg(HuC:GCaMP6s)* larvae. Uncaging was carried out at the desired regions by using 405 nm laser under a Zeiss LSM700 confocal microscope, according to that described previously[32]. Afterwards, Ca[2+] signals were imaged to evaluate the effects.

## 3D reconstruction, photo-conversion, and in vivo time-lapse imaging

The 3D reconstruction of selected images and cell tracking analysis were performed by the Imaris software. To quantify the cell size, the surface renderings of each Z-stack image of coro1a-Dsred[+] cells or IBA1[+] cells were created by Imaris software. Then surface-rendered images were used to calculate the 3D volume. The coro1a-Kaede[+] PM cells in 52 hpf zebrafish larval optic tectum (OT) were selected with the ROI mode and stimulated via a 405 nm laser. The colonization and number of red Kaede[+] cells were quantified at 4 dpf. In vivo time-lapse imaging was carried out according to a previous report[49]. Briefly, the embryos were anesthetized in 0.01% tricaine (Sigma-Aldrich, MS-222) and mounted in 1% low-melting agarose (Thermo Fisher Scientific,

16520050). Then, the midbrain was imaged under an LSM700 confocal microscope with ×20 objective. Images were processed with ZEN or ImageJ software. The calcium signals imaging was performed on *Tg(HuC:GCaMP6s)*. The Annexin V-Cy5 (Biovision, K103) was loaded to the brain to label the apoptotic cells[30]. Acquisitions were performed by scanning at 1-2 Hz at an excitation of 488 nm[58]. The resolution was designed by 256 ×256 pixels on *Tg(HuC:GCaMP6s)*. Imaging data were analyzed by using the ImageJ software and the drift fluorescence images were aligned by StackReg software[59]. The fluorescence intensity of each calcium event in the regions of interest (ROIs) was calculated by (F-F0)/F0. F0 was the average intensity of the ROI accordingly[60]. A spontaneous calcium event was composed of varied calcium transients[61]. The calcium transient with the amplitude exceeding the baseline noise was counted. Only one neuronal calcium event showed nine transients during the imaging time, which was set as the standard to classify the higher and lower frequencies of calcium events. Based on the ΔF/F0, we defined the max amplitude of calcium transients in each event as the peak amplitude[58].

### Single-cell RNA-sequencing (scRNA-seq)

The collected coro1a-DsRed[+] cells were loaded onto the channels of Single Cell G Chip (v3.1 chemistry, PN-1000120). The 10x libraries were constructed using Chromium single-cell gene expression platform and Chromium Single Cell 3′ Reagent Kits (v3.1 chemistry PN-1000121) by AccuraMed Medical Technology Co., LTD (Shanghai, China). Briefly, single-cell suspensions in each channel of the chip were loaded on a Chromium Controller (10x Genomics, Pleasanton, CA) to generate single-cell GEMs (gel beads in emulsion). Then the 10x libraries of each channel were prepared using the Chromium Single Cell 3′Gel Bead and Library Kit v3.1 (PN-1 −1000123, 1000157, 1000129; 10x Genomics). Libraries were sequenced by the Illumina NovaSeq 6000 S4 Reagent Kit v1.5 (illumine PN-20028312).

### scRNA-seq data analysis

The scRNA-seq data were aligned to the zebrafish reference genome (*Danio rerio*, GRCz11) by Cell Ranger 5.0.0 (https://support.10xgenomics.com/single-cell-gene-expression/software/pipelines/latest/what-is-cell-ranger). The resulting matrix was inputted into Seurat v3 for quality control, filtration, dimensionality reduction, clustering analysis and differential expressed genes analysis[62]. In brief, the cells with 200 <nFeature_RNA < 6,000 and percent.mito <30% were obtained for downstream analysis. *Cellcyclesorting* and *ScaleData* were used to eliminate the effects of cell cycle heterogeneity[63]. After identifying the significantly principal components by conducting principal component analysis (PCA), the first 20 principal components were used to compute the k.param nearest neighbors and generated cell clusters. Non-linear dimensional reduction (UMAP) was followed to visualize clustering results. *FindAllMarkers* (*min.pct* = 0.1, *logfc.threshold* = 0.25, *test.use* = "wilcox" (Wilcoxon Rank Sum test)) was employed to filtrate differentially expressed genes (DEGs) across each cluster. Immunocytes, microglia and cluster specific/focused genes were visualized using *Featureplot* and *Dotplot*. Then DAVID Bioinformatics Resources 6.8 were engaged to explain the gene ontology-biological processes[64,65]. The summary terms were listed.

### Bulk RNA Sequencing and data analysis

The cell suspensions were collected from zebrafish larval brains according to the protocols previously described[66]. About 500 HuC-GFP[+] neurons and 100 photo-converted red coro1a-Kaede[+] cells in brains were sorted by flow cytometry (Moflo XDP, Beckman) to establish the cDNA libraries by Anoroda corporation. The RNA sequencing was carried out by using the PE100 strategy (HiSeq 2500, Illumina) and the data were aligned to the zebrafish reference genome (*Danio rerio*, GRCz11). Then, genes with P.adjust <0.1 and log2FoldChange > 0 were identified as differentially expressed genes (DEGs)

using the DESeq (v1.32.0) software. Finally, DAVID Bioinformatics Resources 6.8 were engaged to explain the gene ontology-biological processes of these DEGs[64,65].

### Transplantation

The cell transplantation was performed according to a previous description[67]. The donor brains of 55 hpf WT and *bax[cqSS]/Tg(HuC:GFP)* larvae were dissected according to the method mentioned above. WT and *bax[cqSS]* GFP[+] cells were sorted by flow cytometry. After briefly centrifuging (1,000 x g) and washing, the donor HuC-GFP[+] cells were resuspended with 5% FBS and injected into 55 hpf *bax[cqSS]* recipient larval midbrains by using a micromanipulator (Eppendorf, 5196000030). GFP[+] signals were examined to estimate the transplantation efficiency after 17 h post transplantation (hpt).

### Bacterial phagocytosis

The DsRed labeled *E. coli* (DH5a) bacteria[30] were injected into the midbrain of 2 dpf *Tg(mfap4:GFP)*, 3 dpf *Tg(mfap4:GFP)* and 3 dpf *bax[cqSS]/Tg(mfap4:GFP)* fish. 30 min later, the larvae were anesthetized and mounted in 1% low-melting agarose. Then the fluorescence of bacteria and mfap4-GFP[+] cells was captured by a Zeiss LSM700 confocal microscope. Imaris software was used to quantify the bacterial volume in the mfap4-GFP[+] cells.

### Vector construction and transgenic lines generation

The *coro1a, HuC, NBT, mfap4* and *hsp70* regulatory elements were used. The 7 kb sequence of *coro1a* promoter[31], 3.2 kb sequence of *HuC* promoter[68], 3.8 kb sequence of *NBT* promoter[30], and 1.6 kb sequence of *mfap4* promoter elements[50] were amplified according to previous studies respectively. These promoters were cloned to the *PT2AL2* backbone for further utilization. The coding sequences of *bax, dlb*, zebrafish *notch1a* intracellular domain (z*NICD1a*) and *creb1a* were amplified by the corresponding primers that are listed in Supplementary Table 1. To construct the *coro1a:DN-MAML-FLAG* plasmid, the coding sequences for amino acids 13-74 of murine MAML (dominant-negative mastermind-like 1; DN-MAML) were obtained from *Tg(hsp70:DN-MAML-GFP)*[53] by the listed primers in Supplementary Table 1. This element was then fused to the FLAG epitope at the N terminus. All the successfully amplified elements were then cloned to the *pT2AL2* vectors that contained several tissue-specific regulatory elements and the eye-marker cassette, *Cryaa-Cerulean*[49], was reversely inserted to facilitate screening. *pT2AL2-hsp70-bax* was constructed by sub-cloning the coding sequence of *bax* downstream of the zebrafish *hsp70* promoter[53] and a *Cryaa-Cerulean* was reversely inserted at the same time. The successfully constructed vectors (30 pg) were mixed with Tol2 transposase mRNA (250 pg) and injected into one-cell embryos to perform the transient rescue assays. Transgenic lines *Tg(HuC:GFP)*, *Tg(HuC:dlb)*, *Tg(HuC:bax)*, *Tg(HuC:NICD1a)*, *Tg(NBT:DenNTR)*, *Tg(NBT:creb1a)*, *Tg(hsp70:bax)*, *Tg(coro1a:NICD1a)*, *Tg(mfap4:NICD1a)* and *Tg(coro1a:DN-MAML-FLAG)* were generated in this study. To acquire these stable lines, the constructs, together with transposase mRNA, were injected into one-cell stage embryos. The treated embryos were selected by the transient fluorescence signals and raised into the adulthood F0 founder. The stable hereditary transgenic lines were identified in the F1 generation based on the genetic sequence and the fluorescence signals. The 3 dpf WT or *bax[cqSS]/Tg(hsp70:bax)* and 4 dpf WT *Tg(hsp70:DN-MAML-GFP)* was identified by the cerulean[+] (eyes) and AO[+] (brain) signals after heat-shock at 37 °C for 30 min.

### Chemical, mRNA, and morpholinos (MOs) treatments

The embryos were treated at designed time points by using chemicals: Nemadipine-A (10 μM, 2.5 dpf, 12 h; Santa Cruz, sc-202727); Nilvadipine (40 μM, 2.5dpf, 12 h; Selleck, S2721); DAPT (50 μM, 2.5 dpf, 12 h; Sigma-Aldrich, D5942); KN-92 phosphate (15 μM, 2.5 dpf, 12 h;

ApexBio, A3531); KN-93 phosphate (15 μM, 2.5 dpf, 12 h; Selleck, S7423); PF-CBP1 HCl (35 μM, 2.5 dpf, 12 h; Selleck, S8180); Forskolin (35 μM, 2.5 dpf, 12 h; Selleck, S2449). The apyrase (100 U/ml; Sigma-Aldrich, A6410), ATP (100 mM; Sigma-Aldrich, A6559), carbenoxolone (CBX, 25 μM; Sigma-Aldrich, C4790), Probenecid (70 mM; Sigma-Aldrich, P8761), and PAC-1 (35 μM; Tocris Bioscience, 2581/10) were mixed with Alexa Fluor 647 Dextran (1:1000) and injected into the OT region of 2.5 dpf or 3 dpf anesthetized WT zebrafish embryos. To induce neuronal cell ablation of zebrafish midbrain, 2.5 dpf *Tg*(*NBT:DenNTR*) embryos were treated with metronidazole (MTZ, 15 mM; Sigma-Aldrich, M3761) for 12 h. Then these embryos were rinsed in E3 buffer and analyzed further. The mRNAs of Zebrafish (Z)-*bax* (mut), Zebrafish (Z)-*bax* (WT), Mouse (M)-*bax*, Human (H)-*bax* and *creb1a* were synthesized in vitro through the SP6 kit (Thermo Fisher Scientific, AM1340) by the primers (seen in Supplementary Table 1). The *dlb* MO and *notch1a* MO were used following previous reports[69,70]. The *creb1a* morpholino targets and its control (seen in Supplementary Table 1) were designed by Gene Tools LLC. Both mRNA and MOs were injected into the fertilized embryos at one-cell stage.

## Bax monoclonal antibody production

A monoclonal antibody against zebrafish Bax antigen was generated by Chongqing Biomean Technology Co., Ltd company. Briefly, the designed appropriate Bax antigen segments "GAALLNNFVYERVRRH GDRDAEVTRSQLGGVELCDPSHKRAQCLQQIGDELDGNAQLQSMLNN SNLQPTQDVFIRVAREIFSDGKFN (20-108)" were recombinantly expressed. Next, combined with complete Freund's adjuvant (CFA) and incomplete Freund's adjuvant (IFA), Bax antigen was sub-cutaneously injected into Balb/c mice. The high titer mice were selected after immunized for 1 month or 2 months. Subsequently, we collected the spleen cells of immunized mice to fuse with SP2/0 myeloma cells. The high-affinity and specific anti-Bax antibody strains, such as 3E10, were then screened to infect mice to produce ascites. Finally, the ascites was concentrated and purified with protein A/G. The purified antibody was anti-Bax monoclonal antibody at 3.3 mg/ml. To make an evaluation, WB of Bax antibody (1:400) was performed on the total protein of 12 hpf zebrafish brain after injection of *bax* mRNA.

## Western blotting (WB)

The total proteins of 3 dpf zebrafish brain or E14.5 mouse mesence-phalon were collected to perform WB. The antibodies utilized in this study: mouse anti-zBax, rabbit anti-CaMKII (pan), rabbit anti-phospho-CaMKII (Thr286), rabbit anti-CREB, rabbit anti-CREB (phospho S133), rabbit anti-NICD1a, rabbit anti-activated Notch1, mouse anti-Hes1, mouse anti-GAPDH. To detect the protein level of dlb or DLL3 in the plasma membrane of 3 dpf zebrafish brain or E14.5 mouse mesence-phalon, the membrane was firstly isolated by Minute™ Plasma Mem-brane Protein Isolation and Cell Fractionation Kit (Invent, SM-005). Mouse anti-β tubulin and rabbit anti-Dll3 were then utilized to perform WB. Detections were completed by Goat anti-mouse or anti-rabbit HRP-conjugated secondary antibodies and enhanced chemilumines-cent (ECL Plus) reagent. Each protein sample contains approximately 100 zebrafish brains or 2 mice mesencephalon. Each WB result was repeated three times independently with similar results.

## Validation of the NICD1a, DLL3, p-CaMKII, and p-CREB antibodies

WB was conducted to validate the NICD1a antibody in zebrafish. A specific fragment of approximately 80 kDa was appeared, matching the predicted molecular weight. Further, WT zebrafish embryos were treated with DAPT from 52 hpf to 3 dpf. The total protein of treated zebrafish brains was collected to perform WB. The data indicated an obvious reduction in NICD1a expression in DAPT group. To validate whether the DLL3 antibody authentically binds DeltaB in zebrafish, *pCS2-dla-flag*, *pCS2-dlb-flag*, *pCS2-dlc-flag* and *pCS2-dld-flag* plasmids

were constructed. The primers were listed in Supplementary Table 1. These four plasmids and *pCS2* empty vector (control) were transfected into HEK293T cell lines with Lipo8000 (beyotime, C0533). Two days later, the cells were collected. The cellular proteins of the cultured cells were harvested to perform WB by using DLL3 and FLAG antibodies. The concurrent detection of the bands of 70 kDa, the weight of DeltaB (*pCS2-dlB-flag*), by both FLAG and DLL3 antibody validated the speci-ficity of DLL3 to DeltaB. The molecular sizes of the presumed other Delta proteins are: 87 kDa (*pCS2-dla-flag*), 74 kDa (*pCS2-dlc-flag*), and 81 kDa (*pCS2-dld-flag*). p-CaMKII and p-CREB antibodies were similarly validated. The total proteins of 3 dpf zebrafish brains were immuno-precipitated (IP) with p-CaMKII and p-CREB, respectively. WB assays indicated that the weight of protein labeled by p-CaMKII antibody was the predicted molecular weight (60 kDa) and the weight of protein labeled by p-CREB was approximate 43 kDa of the predicted volume.

## ChIP and luciferase reporter assay

Chromatin immunoprecipitation (ChIP) assay was performed as described previously[67]. 3dpf WT zebrafish DNA was extracted by chromatin immunoprecipitation kit (Sigma-Aldrich, 17-295) and then precipitated by immunoglobulin G (IgG as controls) (Thermo Fisher Scientific, 31460) and p-CREB antibodies. PCR was performed by using the designed primers (see Supplementary Table 1) that targeted the predicted CREB binding sites. The luciferase assays were performed by using the Dual-Luciferase Reporter Assay System as previously described. In brief, luciferase reporter constructs (PGL3) containing WT (site1: TGACGTCA; site2: GACATCA) and mutant (deletion) binding sites in the *dlb* promoter regions were generated. Then together with *pCS2-creb1a* (160 ng) and *pRL*-CMV (40 ng) plasmid, the *PGL3-dlb* (WT, 200 ng) or *PGL3-dlb* (MU1, 200 ng), *PGL3-dlb* (MU2, 200 ng), *PGL3-dlb* (MU1 + 2, 200 ng) plasmids were transfected into HEK293T cell lines via Lipo8000. In this assay, PGL3-basic plasmid was used as a control. After cultured for 2 days, the cells were lysed and the supernatant was collected after centrifugation at $12,000 \times g$. Finally, the supernatant was used to determine the fluorescence intensity.

## Mouse microglia isolation and culture

Primary microglia were directly isolated from C57BL/6J E12.5 embryonic mice. After perfused with ice-cold PBS, mouse brains were rapidly dissected and placed into ice-cold dPBS. Cerebella and spine cords were discarded. The brains were minced to ~1 mm$^3$ with scissors under a dissecting microscope and digested with papain at 37 °C for around 40 min. Then, the digested cell pellets were centrifuged with 800 rpm at 4 °C for 5 min, washed by HBSS, resuspended in 5 mL 5% FBS and passed through 1.2 μm and 0.5 μm glass. Finally, cell sus-pension was pushed through a 40 μm cell strainer to gain the single-cell suspension. Cell suspensions were incubated with FITC anti-mouse/human CD11b antibody (Elabscience, E-AB-F1081C, 5:100 in blocking buffer containing 5% FBS in dPBS) at 4 °C for 2 h. Cell sorting was performed by flow cytometry. Sorted CD11b$^+$ cells were cultured at 24-well plate ($6 \times 10^4$ cells per well) coated with poly-D-lysine, and grown in microglia culture medium containing 10% FBS, 100 U/ml penicillin, 100 μg/ml streptomycin, 10 ng/ml recombinant mouse MCSF (R&D Systems, 416-ML-010/CF) with or without the presence of 1 μg/ml recombinant human DLL3 (R&D Systems, 9749-DL-050) for 5 days.

## Immunofluorescence staining of primary microglia

After a briefly washing with PBS, the primary microglia were fixed in 4% PFA for 10 min at room temperature and washed with PBS. Then the cells were treated with 0.2% Triton X-100 for 15 min, blocked with 5% FBS for 30 min, and incubated with primary antibodies, including goat anti-IBA1, mouse anti-Hes1, rabbit anti-activated Caspase-3, and mouse anti-PCNA (Sigma-Aldrich, P8825, 1:800), at 4 °C overnight. The next day, the cells were washed with PBS for 3 times and incubated with

Alexa Fluor 488 donkey anti-rabbit IgG, Alexa Fluor 647 donkey anti-rabbit IgG and Alexa Fluor 555 donkey anti-mouse IgG at room temperature for 2 h. DAPI (Sigma-Aldrich, 28718-90-3) was used to stain the nucleus. Immunofluorescence staining was imaged under LSM880 confocal microscope.

## Quantitative RT-PCR

Zebrafish and mouse embryonic quantitative RT-PCR (qPCR) were conducted according to a previous method[49]. Zebrafish embryonic total RNA isolation (Roche, 11667165001) and cDNA synthesis (Thermo Fisher Scientific, 18080051) were performed as previous described[49]. HuC-GFP[+] and coro1a-DsRed[+] cells were sorted from 3 dpf zebrafish midbrains by flow cytometry. Mouse mesencephalon cell suspensions were incubated with FITC anti-mouse/human CD11b antibody (Elabscience, E-AB-F1081C, 5:100 in blocking buffer containing 5% FBS in dPBS) and Alexa Fluor 700 anti-mouse CD45 antibody (BioLegend, 103128, 2:100 in blocking buffer containing 5% FBS in dPBS) at 4 °C for 1 h. Then, CD11b[hi]CD45[lo] cells were sorted by flow cytometry. These harvested cells and primary microglial cells were applied to generate cDNA libraries by using REPLI-g WTA Single Cell Kit (QIAGEN, 150063). Relative mRNA levels were determined using the Eppendorf qPCR machine and *gapdh* or *ef1α* was used as the control reference. Samples were run in duplicate or triplicate, and the data are presented as mean ± SEM. Genes primers for qPCR assay were listed in Supplementary Table 1.

## Statistics and reproducibility

The positive signals were manually scored and double confirmed blindly. The quantified data were analyzed by GraphPad Prism 8.3.0. Statistical analysis was performed using unpaired two-tailed Student's *t* test. All error bars represented mean ± SEM. All "n" and "*P*" values and statistical tests were indicated in figures and corresponding legends. The experiments in this study, including live-imaging and uncaging, immunofluorescent and neutral red staining, transplantation, WISH and FISH, drugs treatment, WB and CHIP PCR, luciferase and apoptosis detection, were independently repeated at least three times. Similar results were obtained.

## Reporting summary

Further information on research design is available in the Nature Research Reporting Summary linked to this article.

## Data availability

The RNA-seq and scRNA-seq data had been deposited in the Sequence Read Archive under accession code PRJNA660936 (RNA-seq of H-G[+] neurons), PRJNA661142 (RNA-seq of photo-converted red c-K[+] microglia) and Gene Expression Omnibus under accession code GSE183340 (scRNA-seq), respectively. Previous published scRNA-seq that were re-analyzed here were available under accession codes GSE121654 [45]. Raw data were provided with this paper. Source data were provided as a Source Data file with this paper. All data that support the findings of this study, including experimental animals, antibodies, vector constructions, and chemicals, are available within the article, its Supplementary Information, or from the corresponding author upon reasonable request. Source data are provided with this paper.

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

## Acknowledgements

The authors thank Z. Wen and J. Du for plasmids, antibody and fish lines discussions, H. Huang, H. Li, and D. Wang for technical assistances. This work was supported by National Key Research and Development Project (2019YFA802703); National Natural Science Foundation of China Grants (32270873, 32000568, 31822033); Natural Science Foundation of Chongqing (cstc2020jcyj-msxmX0104), and Fundamental Research Funds for the Central Universities Grant (XDJK2020C041).

## Author contributions

F.Z., J.H., J.T., L.Li, and L.Luo designed the experiments. F.Z., J.H., and J.T. designed and conducted most experiments. L.Luo performed the ENU mutagenesis. Y.S. conducted the mutant screening. N.C., Z.L., and S.L. performed mouse microglia isolation and culture. Y.W., C.Z., M.M., and L.Li provided technical assistance and discussed the results, F.Z. and L.Li wrote the manuscript. All authors read and approved the final manuscript.

## Competing interests

The authors declare no competing interests.
