## [Peer Review File · Nature Communications]

Brain milieu induces early microglial maturation through the
BAX-Notch axisREVIEWER COMMENTS

Reviewer #1 (Remarks to the Author):

I am an expert in microglia development, ontogeny, and identity (and not a zebrafish biologist), and will focus on these topics.

Here, Zhao and colleagues use mainly zebrafish to study interactions between primitive macrophages and neurons early in embryonic development. Their major claim is that a neuronal bax-CamkII-CREB-notch signaling axis confers microglial identity, and the central concern (elaborated below) is that while the authors claim to be studying the regulation of microglial identity, their data more accurately describes one early step in the developmental maturation of microglia, and the paper would be much stronger if this were the focus.

The authors first characterize in zebrafish a change in macrophage phenotype that occurs soon after their arrival in the developing CNS that seems to rely on cell-cell contact between microglia and neurons. This phenotype change is manifest in morphology, migration speed, and expression of some microglia specific genes. They go on to show that this phenotype change, which I think could best be described as a potential "maturation step" in microglia development, requires neuronal bax, is mediated by notch signaling, requires neuronal firing, ATP, and CamKII-CREB signaling in neurons. Finally, they perform experiments in mouse to study whether 1) bax/notch are necessary for normal microglial numbers, morphology, and gene expression and 2) whether Dll3, a notch ligand, is sufficient to induce microglial gene expression in vitro.

Overall, the data in this manuscript are new, important, and have potential to be widely appealing, but its presentation, interpretation and the current conclusions greatly limit impact.

In terms of presentation, I found the text and figures difficult to read and follow. As one example of many: figure 1 panels show results from the cq55 mutant, but the text does not mention these results until long after the reader sees them in the figure, which made figure 1 hard to follow as results were described in the text out of order from how they appeared in the figure. The text would also benefit from comprehensive editing for clarity and precision of language (specific examples from one results section are provided in "other points" below but the comment applies throughout.)

In terms of content, the major concern for the paper is that the authors appear to be telling a story about one early step in the process of microglial maturation, rather than the regulation of identity, and I would strongly recommend that they modify their claims accordingly. To elaborate, the authors frame their results around "microglia identity," claiming that the early developmental interaction they have identified is a major regulator of identity. Microglial identity is not an explicitly defined concept in the field, so first off it's not clear exactly what the authors are claiming. To me, at the very least, proving that early developmental notch signaling regulates microglial identity would require showing that specific absence of this interaction during development leads to a long lasting and epigenetically regulated change in the ability of microglia to express microglial genes (and ideally, perform microglial functions) when exposed to a healthy and mature brain environment. A closely related concept is the idea of brain macrophage state - this is something that non microglial macrophages can also take on in response to environmental signals - for example, monocytes that engraft the brain parenchyma can express most microglial genes, but do not have microglial identity. The main conclusions I draw from the data shown here are not about microglial identity, and the experiments do not disambiguate between an effect on state, versus a durable effect on identity. Thus, the authors here have not substantiated their claims about regulation of microglial identity. In their zebrafish studies, they have identified a very interesting and potentially important step in the developmental programming of microglia. This alone is appealing to a wide audience, important, and warrants publication, and I would recommend making this the focus of the paper.

The above is particularly important in terms of interpretation of data from mouse experiments. In their bax mutant studies, it is not clear whether the authors are studying the analogous developmental

timepoint to their zebrafish work, so not clear if they are looking at the “same” early interactions in mouse (eg at microglial programming), or instead at how sustained notch signaling is important for microglial gene expression (eg, at regulation of microglial state). Similarly, in the Cx3cr1 Cre- Rbpj fl/fl experiments (KO of an important notch regulator), I cannot find information on animal age. Finally, the in vitro experiments in figure 8 are performed with adult microglia, eg cells that already have attained microglial identity. In addition to the effects of DLL3 on microglial gene expression being of very small magnitude in these experiments, to me this seems to be testing something very different than the developmental programming studied in prior experiments, and so its relationship to other claims in the paper is also not clear.

Other points:

1. Mouse FACS plots to quantify microglia - The plots shown are difficult to interpret - without a gating strategy, as well as the standard CD45/11B plots canonically used to identify microglia. This is important because in the plots shown in 7E and 8C, at face value at least the authors are gating on “f4/80 high” cells rather than “f4/80 low” cells, which are more likely to be the microglia (or, the “f4/80 low” population I am referring to is actually debris that’s not gated out, but then this becomes even more confusing to interpret)

2. In Figure 8a-h, the authors mention n=5 biological replicates but also that each dot represents 1 cell or 1 slice. I am not sure what this means as there are more than 5 dots in each group - this would suggest that each dot is not an independent replicate, and so cannot be treated as such for statistical inference.

3. For the Cre-fl Rbpj experiments, the Cx3cr1 Cre may have some effects on phenotype. It would increase rigor to show a Cre only control in addition to WT, and to verify Rbpj loss specifically in microglia.

4. The authors compare gene expression in sorted microglia cultured without csf1, to those cultured with csf1, and those cultured with csf1 and dll3. A lack of csf1 would be expected to lead to poor survival of cultured cells - this is not addressed in the manuscript and it’s not clear what the authors are measuring when reporting this condition, as I would predict most cells to be dead or dying.

5. The authors made a new bax antibody, but little information is given about how it was made. Also, especially because the authors have mutants, it is important to show that the antibody does not stain bax KO tissue. This is particularly important because in multiple RNAseq datasets including Hammond/Stevens and Li/Barres, Bax is expressed by microglia at the timepoints studied in mouse.

6. The authors performed a forward genetic screen but no information is presented about the results of the screen besides cq55. I do not have zebrafish expertise, but if this screen is used here, I would expect that the authors should provide information on it such as parameters used to perform it, readouts, how many hits there were, why they chose to focus on bax, etc.

7. Many western blots seem to show a single n - it’s important to convey how many times biochemistry was independently replicated, and somehow quantify whether or not the result was consistent. More broadly, all group sizes should be made clear for all experiments.

8. Specific examples of where text editing could improve clarity and precision, from the results section “Apoptotic neurons engage nucleosides to locate microglia for Notch signaling.” my point is not to prescribe specific edits here, just am giving some examples to clarify the point.

8a. This title could be clarified for eg “Microglial sensing of nucleosides released by apoptotic neurons potentiates notch signaling”

8b. “The c-D+ cells lost adhesion and actively moved away, causing a marked decrement in the c-K+ and apoeb+ populations in the treated brain” - the authors don’t test adhesion, so this could be edited to reflect experiments actually done.

8c. "Synergistic application of ATP and dlb enhanced the efforts in approximate 31.20% of samples than that 21.98% larvae supplying dlb at 2 dpf"

8d. "Altogether, these results indicate that nucleosides (given by apoptotic neurons) properly placed microglia to receive Notch instruction." → "altogether, these results show that neuronal ATP release facilitates proper localization and in turn programming of microglia"

Reviewer #2 (Remarks to the Author):

The manuscript presents a previously unidentified communication between developing neurons and microglial progenitors that controls microglial differentiation. This communication is defined in remarkable detail through well-designed genetic and pharmacologic experiments in zebrafish and mice. The manuscript demonstrates that BAX protein expressed in neurons controls neuronal signaling to microglia, that NOTCH-DELTA signaling mediates this communication. They show that DELTA expression by neurons, modulated BAX-dependent calcium currents, activates NOTCH signaling in microglia. Overall, the paper presents a new concept in development, but key aspects of the neural-microglial communication are not fully new. In particular, signaling of microglia through BAX-dependent neuronal apoptosis has been described (for example, Anderson et al Cell Reports 2019 PMID 31091440) and NOTCH signaling in microglia has been described (for example, Wei et al Stroke 2011, PMID 21737799). While this paper finds similar mechanisms through their own independent approach, they should discuss their findings in the context of these and other relevant works, and delineate what is new and what is not. There are additionally weaknesses in the work which must be addressed, but it is overall a very strong story and it should be reported in Nature Communications.

The writing and grammar are poor throughout the text, and the linguistic issues are distracting. The authors have many details to report, and the task for the reader is complicated when the reader has to guess what the author is saying. Microglia should not be described as "strolling" or "lethargic" as these terms are not clear or free of ambiguity in the context of cells in the brain. Extensive communication between the authors and a copy editor are needed to make sure that textual revisions do not alter the intended meaning, but make the meaning more clear.

A more thematic concern regards whether or not apoptosis per se a mechanism of BAX-dependent regulation of microglial differentiation. The language of the paper at times seems to argue that apoptosis is not a determinant of microglial identity, and this conclusion does not seem consistent with the data. Indeed, the discussion seems to include neuronal apoptosis as a direct regulator of microglia which would make more sense. The issue may be more with communication and writing than with intended content, but it needs to be addressed.

The data clearly show that apoptosis is not the only mechanism through which BAX alters microglial development, as inducing apoptosis with the NTR-Mtz system or PAC-1 did not rescue apoE expression. However, these data do not show that apoptosis does not influence microglia, but rather only that apoptosis alone is insufficient to fully restore neuron-microglial communication. Indeed in Fig 3A, Mtz seems to have induced some increase in mpeg1 expressing cells in Bax mutants. In contrast, the data on nucleoside release altering microglia clearly show that apoptotic cells do affect the microglial population. A more persuasive conclusion, consistent with all of the data, would be that BAX-dependent processes in neuronal cells influence microglia through both apoptotic and non-apoptotic mechanisms that are molecularly distinct. The authors should consider whether this explicit phrasing is acceptable and adds clarity.

Specific text issues (not including grammatical and word choice issues which are too numerous to list):

The text should briefly explain what is meant by cq55 mutant.

The text should briefly explain the NTR-metronidazole system.

The abbreviation AO+ is not defined before it is used.

Line 257 the sentence “Uncaged IP3 or glutamate was injected into baxcq55 midbrains” should read “Cage IP3 or glutamate was injected into baxcq55 midbrains”

Fig 1K- The authors report rich factor, which is understood to be the fraction of genes in a GO pathway that are differentially expressed in the sample. Is it correct that 100% of the genes in the isocitrate dehydrogenase pathway were differentially expressed?

In Fig 3A, MTZ seems to increase mpeg1+ cells in the Bax mutant. Please provide statistical analysis and if mpeg1+ cells are increased, discuss the divergence of mpeg1 and apoe data. Also, should mpeg1 be analyzed in other mutants, such as dll KO?

Reviewer #3 (Remarks to the Author):

In this manuscript, Zhao et al identify neuronal Delta/Dlb as a novel signal that regulates microglia differentiation. The authors rigorously show that Delta is induced downstream of neuronal activity in a Bax/CAMKII/CREB-dependent manner. Thus, the authors uncouple the role of Bax in promoting neuronal apoptosis— which initially attracts microglia to the nervous system— from a newly identified role in cell signaling, which promotes microglia maturation after their arrival. This is an impressive body of work with impactful findings, since the brain-specific cues that specify and maintain microglia identity are currently not well understood. However, this manuscript requires extensive editing for grammar and syntax as it is currently difficult to understand, making the findings more challenging to appreciate. Following a few revisions (listed below), this study should be published with no delay.

Major Comments

1. The authors conclude that neuronal Delta signals directly through microglial Notch to regulate microglial development. However, direct evidence that Delta is acting through Notch on microglia is not provided. The authors should perform epistasis experiments to support this claim— for example:
 - Inhibit microglial Notch while rescuing neuronal delta in the Bax mutant, or
 - Inhibit microglial Notch while uncaging Ca²⁺
2. Line 63-65: The authors state that maturing microglia begin to engulf dying cells. However, none of the figures presented demonstrate the engulfment of “gradually appeared Casp3+ apoptotic cells.” The authors should provide images and 3D reconstruction showing this result.
3. The manuscript requires extensive editing for grammar and syntax, including the abstract. For example:
 - Line 22-24: “Bax concomitantly retains the pre-microglia through apoptotic neurons to ensure effective obtain of Notch instructions from the surrounding neurons. Notch signalling in regulating microglia signatures is faithful.” The first sentence should be rewritten, and the word “faithful” should be changed to “conserved.”
 - Line 32: “Accumulating evidences indicate”
 - Line 40: “It was recently suggested that rescue of damaged microglia or generation of safe microglia surrogates from iPSCs could prevent even reverse neurological disease progression.” What does “safe microglia surrogates” mean? Healthy? Homeostatic?
 - Use of “fortuitously” on line 120 and “unfortunately” on line 122 appears emotional and unnecessary.
 - Line 315: “This result followed the failed rescued outcomes of solely increasing apoptotic cells in baxcq55 brain.” This sentence is confusing and should be written, perhaps: “This result is consistent with the observation that solely increasing apoptotic cells in baxcq55 brain failed to rescue the

microglia differentiation phenotype.”

4. The authors should carefully check the relevance of the literature cited to assess whether the findings in these papers actually support their arguments. Some examples where papers were inappropriately cited include:

- Line 35-36: The authors state that NGF is a neurotrophic factor that supports the proliferation of neural progenitor cells, but cite a paper that demonstrate NGF’s role as a killing agent (Frade et. al, 1998).
- Line 37-38: The authors cited a paper that demonstrated microglial functions in mediating forgetting and erasure of memory (Wang et. al, 2020), but the authors used this reference to support an argument that microglia improve learning and memory performance.
- Line 112: The authors state that microglia heterogeneity has been demonstrated in mammals, but cite a paper focusing on differences between CNS border-associated macrophages and microglia (Utz et. al, 2020).

Minor Comments

1. The coloring in Figure 1B is really confusing. It would be better to use only 2 colors, one for “sib” and one for “cq55.” The authors should also define the term “sib” in the text and/or legend.
2. Figure 1H should be presented as an earlier panel, as this information can help readers better understand the labeling used for each cell type.
3. Could the increased contact between microglia and neurons in Figures 1E and 1F be explained by increased neuronal density during development? This factor should be considered and discussed.
4. On line 125, the authors state that “More than half of the cells entered into the eyes, hindbrains, and ventral brains, which caused a drastic Reduction in c-D+ pools,” but Figure 1I-J do not show data supporting this claim.
5. Figure 1L-N are not referenced or explained in the main text. Please describe this experiment. It is unclear what Figure S1F is showing.
6. On line 174-180, the authors list GO terms that are altered in mutants. For clarity, the authors should explicitly state the GO terms enriched for downregulated versus upregulated genes. This should also be made clear in Figure 3C.
7. Figure 1N should have more neurons drawn in the mutant condition.
8. Figure S8G should be included in the main figures.
9. Please introduce acronyms and field-specific terminology when first using them, including in figure legends. Some examples of terms that should be defined are:
 - NR
 - WISH
 - Z-bax, M-bax, H-bax
10. How specific is apoeb as a marker for microglia maturation? Since apoeb is the marker used as the main readout of this study, further background or support for using this marker should be provided.
11. Figure 7M is a great schematic but should not be labeled “mouse.” The authors rigorously demonstrate the role of the schematized pathway in zebrafish, but the mouse data presented is not sufficient to conclude that this pathway as schematized regulates microglia maturation in mice.
12. On line 155, the authors state that “Bax is an indispensable neuronal factor in training microglia.”

The word “training” should be changed, perhaps to “regulating microglia maturation,” since immune cell training describes a specific, separate biological phenomenon similar to microglial priming.

REVIEWER COMMENTS

Reviewer #1 (Remarks to the Author):

I am an expert in microglia development, ontogeny, and identity (and not a zebrafish biologist), and will focus on these topics.

Here, Zhao and colleagues use mainly zebrafish to study interactions between primitive macrophages and neurons early in embryonic development. Their major claim is that a neuronal bax-CamkII-CREB-notch signaling axis confers microglial identity, and the central concern (elaborated below) is that while the authors claim to be studying the regulation of microglial identity, their data more accurately describes one early step in the developmental maturation of microglia, and the paper would be much stronger if this were the focus.

The authors first characterize in zebrafish a change in macrophage phenotype that occurs soon after their arrival in the developing CNS that seems to rely on cell-cell contact between microglia and neurons. This phenotype change is manifest in morphology, migration speed, and expression of some microglia specific genes. They go on to show that this phenotype change, which I think could best be described as a potential “maturation step” in microglia development, requires neuronal bax, is mediated by notch signaling, requires neuronal firing, ATP, and CamKII-CREB signaling in neurons. Finally, they perform experiments in mouse to study whether 1) bax/notch are necessary for normal microglial numbers, morphology, and gene expression and 2) whether Dll3, a notch ligand, is sufficient to induce microglial gene expression in vitro.

Overall, the data in this manuscript are new, important, and have potential to be widely appealing, but its presentation, interpretation and the current conclusions greatly limit impact.

In terms of presentation, I found the text and figures difficult to read and follow. As one example of many: figure 1 panels show results from the cq55 mutant, but the text does not mention these results until long after the reader sees them in the figure, which made figure 1 hard to follow as results were described in the text out of order from how they appeared in the figure. The text would also benefit from comprehensive editing for clarity and precision of language (specific examples from one results section are provided in “other points” below but the comment applies throughout.)

Answer: Many thanks to your careful evaluation and constructive suggestions. We are quite sorry for the language presentation and misleading organization in the previous manuscript. To make the information clear, we substantially re-arranged the story frameworks, especially Figure 1, and polished the manuscript. The regarding description was re-organized and presented in the text accordingly (page 5, 6, 7, line 101-118, 127-137, marked by grey background). We also asked both the English language editing service and a native speaker to make the improvement. The revisions were performed accordingly. Thanks very much again!

In terms of content, the major concern for the paper is that the authors appear to be telling a story about one early step in the process of microglial maturation, rather than the regulation of identity, and I would strongly recommend that they modify their claims accordingly. To elaborate, the authors frame their results around “microglia identity,” claiming that the early developmental interaction they have identified is a major regulator of identity. Microglial identity is not an explicitly defined concept in the field, so first off it’s not clear exactly what the authors are claiming. To me, at the very least, proving that early developmental notch signaling regulates microglial identity would require showing that specific absence of this interaction during development leads to a long lasting and epigenetically regulated change in the ability of microglia to express microglial genes (and ideally, perform microglial functions) when exposed to a healthy and mature brain environment. A closely related concept is the idea of brain macrophage state - this is something that non microglial macrophages can also take on in response to environmental signals - for example, monocytes that engraft the brain parenchyma can express most microglial genes, but do not have microglial identity. The main conclusions I draw from the data shown here are not about microglial identity, and the experiments do not disambiguate between an effect on state, versus a durable effect on identity. Thus, the authors here have not substantiated their claims about regulation of microglial identity. In their zebrafish studies, they have identified a very interesting and potentially important step in the developmental programming of microglia. This alone is appealing to a wide audience, important, and warrants publication, and I would recommend making this the focus of the paper.

Answer: We appreciate the careful assessment and acknowledge the instrumental comments. We apologize for the inaccurate statement of microglial identity in previous manuscript. As mentioned, we have identified a potentially important step in the developmental programming of zebrafish microglia. This phenotype change could be better defined as one early event in the microglia developmental maturation rather than microglial identity. We corrected the claim from “microglial identity” to “microglial maturation” in the revised manuscript. The title was changed to “Brain milieu induces microglia maturation through the BAX-Notch axis” accordingly. The regarding description was included. Thank you very much again for the kind suggestion and valuable advices!

The above is particularly important in terms of interpretation of data from mouse experiments. In their bax mutant studies, it is not clear whether the authors are studying the analogous developmental time point to their zebrafish work, so not clear if they are looking at the “same” early interactions in mouse (eg at microglial programming), or instead at how sustained notch signaling is important for microglial gene expression (eg, at regulation of microglial state). Similarly, in the Cx3cr1 Cre- Rbpj fl/fl experiments (KO of an important notch regulator), I cannot find information on animal age. Finally, the in vitro experiments in figure 8 are performed with adult microglia, eg cells that already have attained microglial identity. In addition to the effects of DLL3 on microglial gene expression being of very small magnitude in these experiments, to me this seems to be testing something very different than the developmental programming studied in prior experiments, and so its relationship to other claims in the paper is also not clear.

Answer: We appreciate the constructive comments and suggestions. (a) In zebrafish, embryonic macrophages invade the cephalic mesenchyme from the yolk sac at 22-40 hour post fertilization (hpf) and undergo a phenotypic transition to early microglia at 55-60 hpf¹. These early microglia are mainly

restricted to the optic tectum, especially the neuronal soma layer, at 60-84 hpf^{1, 2}. Larval zebrafish microglia show a rapid differentiation which is reflected by the acquisition of the core microglia signature and the strong upregulation of many microglia specific genes from 3 dpf onwards³. In the mouse, microglial progenitors arise early from the embryonic yolk sac and immigrate into the embryonic brain parenchyma from around E9.5-E10 until the formation of the blood-brain barrier (BBB) at E13.5-E14.5⁴. In addition, mouse microglia development has been proposed to progress in three developmental stages: early microglia (until E14), pre-microglia (from E14 to the first weeks after birth), and adult microglia (from a few weeks after birth onward)⁵. Pre-microglia express some genes associated with neural maturation and synaptic pruning. Based on this information, we chose 3 dpf zebrafish and E14.5 mouse embryos to characterize one early step in the process of microglial maturation in the revised manuscript. The related background introductions were included (page 3, 4, line 45-47, 52-54, 64-68, marked by yellow background). (b) We have added the age information of the *Cx3cr1^{cre/+}Rbpj^{fl/fl}* mice in the corresponding figures and figure legends. (c) We are sorry for the improper developmental time point of mice chosen to perform the *in vitro* culturing assay in previous manuscript. We realized that the signatures of adult microglia were different to that of embryonic counterparts. To validate the functions of Notch signaling in embryonic microglia maturation, we isolated E12.5 embryonic microglia and furnished them with DLL3. The result indicated an improved maintenance of microglia survival and increased expression of microglial homeostatic signature genes (MHSGs) in the medium containing DLL3 and MCSF than that added MCSF only. The data was presented in the revised Figures (Fig. 8f-i, Supplementary Fig. 10a-e) and the description was included accordingly (page 18, 19, line 404-415, marked by turquoise background)

Other points:

1. Mouse FACS plots to quantify microglia - The plots shown are difficult to interpret - without a gating strategy, as well as the standard CD45/11B plots canonically used to identify microglia. This is important because in the plots shown in 7E and 8C, at face value at least the authors are gating on “f4/80 high” cells rather than “f4/80 low” cells, which are more likely to be the microglia (or, the “f4/80 low” population I am referring to is actually debris that’s not gated out, but then this becomes even more confusing to interpret)

Answer: We apologize for lacking a gating strategy and using improper markers to quantify mouse microglia in previous manuscript. We gratefully appreciate your careful assessment and acknowledge the instrumental suggestion. We realized that CD11b^{high} and CD45^{low} were canonically used to identify microglia in mice⁶. In the revised manuscript, we firstly removed the FSC-A^{low} and SSC-A^{low} debris according to a previous study⁷. We then focused on the population of CD11b^{hi}CD45^{lo} cells in mouse mesencephalon by using the gating strategies. The results indicated that the percentage of CD11b^{hi}CD45^{lo} microglia was reduced in E14.5 *Bax^{tm1Sjk}* and *Cx3cr1^{cre/+}Rbpj^{fl/fl}* mice when compared with their sibling controls. The FACS plots and corresponding information were included in the revised manuscript (Supplementary Fig. 8e, 8f, 9f, and 9i, page 16, 17, 18, line 362-365, 396-398, marked by green background).

2. In Figure 8a-h, the authors mention n=5 biological replicates but also that each dot represents 1 cell or

1 slice. I am not sure what this means as there are more than 5 dots in each group - this would suggest that each dot is not an independent replicate, and so cannot be treated as such for statistical inference.

Answer: We sincerely appreciate the precise estimation and apologize for the unclear introduction of quantification numbers in the previous manuscript. We clarified the information regarding sample sizes/animal numbers in the revised figure legends. For example, each dot in the current Fig. 7d, 8b, and Supplementary Fig. 9b, 9l (quantification of fluorescent intensity emission values (RBPJ and Hes1) or the volumes of IBA1⁺ cells) represents individual microglial cell. Approximately ten cells per mouse were quantified ($n = 3$ mice in each group). However, each dot in the revised Fig. 7b, 8d, and Supplementary Fig. 8c, 9e (quantification of cell density) represents individual mouse. Three sections per mouse were quantified. Thanks very much again for the careful concern and comments.

3. For the Cre-fl Rbpj experiments, the Cx3cr1 Cre may have some effects on phenotype. It would increase rigor to show a Cre only control in addition to WT, and to verify Rbpj loss specifically in microglia.

Answer: We are grateful to the suggestive advices. To detect whether Cx3cr1-Cre has effects on microglia phenotype in our study, the CD11b^{hi}CD45^{lo} microglia were carefully analyzed in WT and *Cx3cr1*^{Cre/+} mice at the embryonic stages. The data of flow cytometry analysis displayed a comparable percentage of CD11b^{hi}CD45^{lo} cells in the E14.5 brains between *Cx3cr1*^{Cre/+} and WT mice. Similarly, the transcript levels of MHSGs did not present notable alterations in the microglia from *Cx3cr1*^{Cre/+} mice when compared to that in the WT controls. These data suggested the limited influence of Cx3cr1-Cre on embryonic microglia development. However, the *Cx3cr1*^{Cre/+}*Rbpj*^{fl/fl} mice presented impressively notable reduction of IBA1⁺ and F4/80⁺ cells in compared to their counterparts in E14.5 *Cx3cr1*^{Cre/+} or *Rbpj*^{fl/fl} mice. The image data and corresponding information were included in the revised manuscript (Fig. 8c, d, Supplementary Fig. 9d-h, page 18, line 390-396, marked by grey background). Many thanks again!

4. The authors compare gene expression in sorted microglia cultured without csf1, to those cultured with csf1, and those cultured with csf1 and dll3. A lack of csf1 would be expected to lead to poor survival of cultured cells - this is not addressed in the manuscript and it's not clear what the authors are measuring when reporting this condition, as I would predict most cells to be dead or dying.

Answer: We are sorry for the insufficient experiments of cell culture assays. To further evaluate the outcome of different culture conditions as suggested, we sorted CD11b⁺ cells from E12.5 mouse cerebral cortexes and cultured them in a 12-well plate (6×10^4 cells per well in 2 ml). Two days later, the cells were collected to conduct the immunofluorescence staining. The results indicated a notable reduction of active Casp3⁺IBA1⁺ signals but significantly increment of PCNA⁺IBA1⁺ signals when the isolated cells were incubated with MSCF and MCSF+DLL3. These data indicated that lack of MCSF led to poor survival of cultured cells and MCSF or additional DLL3 culture promoted the survival and proliferation of embryonic microglial cells. These data were presented in the revised Supplementary Fig. 10c-e. The information was included in the new text and legends accordingly (page 18, 19, line 404-415, marked by turquoise background). Thanks very much again for the careful concern and suggestive comments.

5. The authors made a new bax antibody, but little information is given about how it was made. Also,

especially because the authors have mutants, it is important to show that the antibody does not stain bax KO tissue. This is particularly important because in multiple RNA-seq datasets including Hammond/Stevens and Li/Barres, Bax is expressed by microglia at the time points studied in mouse.

Answer: We appreciate the helpful assessment and apologize for lacking the information of new Bax antibody. (a) The antibody was produced by the company (Chongqing Biomean Technology Co., Ltd) and the information was given in the revised methods (page 29, line 689-700, marked by blue color). Briefly, the Bax antigen was prepared to immunize the Balb/c mice. The spleen cells of immunized mice were collected and fused with the myeloma cells SP2/0. Next, the positive antibody strains were isolated and used to produce ascites in mice. Finally, Bax antibody was purified from the protein A/G concentrated ascites. (b) To verify the Bax antibody, we performed the immunostaining and of Bax WB in the brain samples from *bax^{cq55}* mutant and siblings (sib). The immunostaining result indicated that the signals of Bax were easily detected in the sib brain sections but obviously diminished in that from the *bax^{cq55}* mutant (Supplementary Fig. 2c). Similarly, WB results presented a remarkable reduction of zBax levels in the *bax* mutant alleles relative to that in siblings (Supplementary Fig. 2d). (c) According to your kind suggestion, we examined the transcription levels of *bax* in the H-G⁺ neuron and c-D⁺ microglial cells by qPCR. The result indicated a significant reduction but still detectable expression of *bax* transcript in c-D⁺ microglia compared to H-G⁺ neuron (Fig. 2e). We then retrospect the original data of Bax immunostaining in c-D⁺ cells. Indeed, a very weak Bax signal was observed in the c-D⁺ microglia, but it was unfortunately neglected by background reducing and contrast enhancing during the data processing process. We deeply apologized for this. To further confirm this result, we repeated the immunostaining assay. The result indicated that Bax was faintly detected in both the zebrafish c-D⁺ (Fig. 2d) and mice F4/80⁺ cells (Fig. 7a). The improved information was included in the revised manuscript (page 7, 16, line 150-151, 359-360, marked by yellow background).

6. The authors performed a forward genetic screen but no information is presented about the results of the screen besides *cq55*. I do not have zebrafish expertise, but if this screen is used here, I would expect that the authors should provide information on it such as parameters used to perform it, readouts, how many hits there were, why they chose to focus on *bax*, etc.

Answer: Thank you very much for your kind suggestion. We carried out a forward genetic screen in a N-ethyl-N-nitrosourea (ENU) mutagenesis zebrafish library⁸ to seek novel factors in microglia development. *apoeb* is a mature microglia marker at embryonic stages and therefore selected to identify the candidate. A total of 203 crossing pairs were examined by performing *apoeb* WISH at 3 dpf. 66 candidate mutant pairs initially presented the obvious phenotypes in *apoeb*⁺ signals in the brain. The double confirm experiments disclosed two putative mutants lacking *apoeb*⁺ signals. The subsequent complementation tests by crossing these two mutants with known microglia defect mutants of *csf1ra*¹, *pu.1*⁹, and *irf8*¹⁰ indicated that *cq55* was probably a novel recessive mutant. We therefore focused on *cq55* for further study. The related information was included in the revised manuscript (page 6, 22-23, line 119-127, 502-517, marked by turquoise background).

7. Many western blots seem to show a single n - it's important to convey how many times biochemistry was independently replicated, and somehow quantify whether or not the result was consistent. More

broadly, all group sizes should be made clear for all experiments.

Answer: We appreciate the careful assessment and kind suggestion. The experiment of western blot was independently repeated three times and the results were consistent. The information of biological replicates and group sizes was included in the revised figure legends accordingly.

8. Specific examples of where text editing could improve clarity and precision, from the results section “Apoptotic neurons engage nucleosides to locate microglia for Notch signaling.” my point is not to prescribe specific edits here, just am giving some examples to clarify the point.

8a. This title could be clarified for eg “Microglial sensing of nucleosides released by apoptotic neurons potentiates notch signaling”

Answer: Thank you very much for your kind comments. We apologize for the previous description in the text. To make it more accessible, we re-organized the manuscript and made a substantial revision. The previous section “Apoptotic neurons engage nucleosides to locate microglia for Notch signaling” was separated into two parts. The corresponding titles of each new section were given accordingly. We hope the clarity is improved in new manuscript.

8b. “The c-D⁺ cells lost adhesion and actively moved away, causing a marked decrement in the c-K⁺ and apoeb⁺ populations in the treated brain” - the authors don’t test adhesion, so this could be edited to reflect experiments actually done.

Answer: We revised the description accordingly (page 9, line 186-188, marked by blue color).

8c. “Synergistic application of ATP and dlb enhanced the efforts in approximate 31.20% of samples than that 21.98% larvae supplying dlb at 2 dpf”

Answer: It was down (page 13, line 272-278, marked by blue color).

8d. “Altogether, these results indicate that nucleosides (given by apoptotic neurons) properly placed microglia to receive Notch instruction.” → “altogether, these results show that neuronal ATP release facilitates proper localization and in turn programming of microglia”

Answer: We made a major revision on the previous manuscript and asked both the English language editing service and a native speaker to polish our manuscript. We gratefully appreciate your great efforts and valuable suggestion again!

Reviewer #2 (Remarks to the Author):

The manuscript presents a previously unidentified communication between developing neurons and microglial progenitors that controls microglial differentiation. This communication is defined in remarkable detail through well-designed genetic and pharmacologic experiments in zebrafish and mice. The manuscript demonstrates that BAX protein expressed in neurons controls neuronal signaling to microglia, that NOTCH-DELTA signaling mediates this communication. They show that DELTA expression by neurons, modulated BAX-dependent calcium currents, activates NOTCH signaling in microglia. Overall, the paper presents a new concept in development, but key aspects of the neural-microglial communication are not fully new. In particular, signaling of microglia through BAX-dependent neuronal apoptosis has been described (for example, Anderson et al Cell Reports 2019 PMID 31091440) and NOTCH signaling in microglia has been described (for example, Wei et al Stroke 2011, PMID 21737799). While this paper finds similar mechanisms through their own independent approach, they should discuss their findings in the context of these and other relevant works, and delineate what is new and what is not. There are additionally weaknesses in the work which must be addressed, but it is overall a very strong story and it should be reported in Nature Communications.

Answer: We appreciate the careful assessment. (a) Anderson et al pioneering work reported that BAX-dependent neuronal apoptosis affected disease-associated microglia (DAM)-related gene expression in postnatal retinal microglia in mice¹¹. Their study documented that the developmental apoptosis was a major driver of the disease-related profile in microglia. However, our present study focused on the mechanism of microglial maturation at embryonic stages. By using the *bax*^{cq55} zebrafish and *Bax*^{tm1Sjk} mice embryos, our data unraveled that BAX-regulated neuronal apoptosis was required but insufficient to promote embryonic microglia maturation. (b) The engagement of Notch signaling in the reactions (polarization and activation) of adult/matured microglia to the inflammation and immunization was reported^{12, 13, 14}. However, whether and how Notch signaling is applied by the brain milieu to dictate nascent microglia maturation was mysterious. Our present study revealed that the Notch signaling, regulated by BAX in living neurons, was an essential factor in this process. BAX harnessed microglia maturation in two ways. BAX properly positioned microglia by its classical pro-apoptotic roles. Simultaneously, BAX equipped Notch ligands in the living neurons, which effectively activated Notch signaling in the contacted microglial cells. We included this information in the revised manuscript (page 4, 20, 21, 22, line 69-83, 439-442, 469-480, mark by grey background). Thanks very much again for the careful concern and comments.

The writing and grammar are poor throughout the text, and the linguistic issues are distracting. The authors have many details to report, and the task for the reader is complicated when the reader has to guess what the author is saying. Microglia should not be described as “strolling” or “lethargic” as these terms are not clear or free of ambiguity in the context of cells in the brain. Extensive communication between the authors and a copy editor are needed to make sure that textual revisions do not alter the intended meaning, but make the meaning more clear.

Answer: Many thanks to your careful evaluation and suggestions. We are sorry for the poor writing and confused presentation in the previous manuscript. We re-organized the data and made a substantial revision. We asked both a native speaker and the English language editing service to improve and polish

the manuscript to make it easily followed. We hope the revised narration is more explicit and accessible.

A more thematic concern regards whether or not apoptosis per se a mechanism of BAX-dependent regulation of microglial differentiation. The language of the paper at times seems to argue that apoptosis is not a determinant of microglial identity, and this conclusion does not seem consistent with the data. Indeed, the discussion seems to include neuronal apoptosis as a direct regulator of microglia which would make more sense. The issue may be more with communication and writing than with intended content, but it needs to be addressed.

Answer: We gratefully appreciate the careful assessment and acknowledged the instrumental comments. We apologize for the confused delineation of neural apoptosis in microglial maturation. Previous studies documented that the nucleosides released by apoptotic neurons played a crucial role in the immigration of microglia into the brain¹⁵. Consistently, we found that inducing neural apoptosis or providing nucleosides promoted the recruitment of macrophages/microglia into the midbrain. However, no recovery of mature microglia was observed simultaneously in the *bax*^{-/-} mutants (Fig. 3a, b, Supplementary Fig. 3d, g). We further elucidate that Notch signaling was indispensably involved. Therefore, BAX controlled neuronal apoptosis ensured pre-microglia to receive Notch signals from living neurons efficiently in their maturation. Both apoptosis and Notch signaling cooperatively promoted embryonic microglia development. We substantially revised the text to make the conclusion more explicit and accessible. The related description was added accordingly (page 9, 21, 22, line 181-183, 469-480, marked by **grey background**). Many thanks again!

The data clearly show that apoptosis is not the only mechanism through which BAX alters microglial development, as inducing apoptosis with the NTR-Mtz system or PAC-1 did not rescue *apoe* expression. However, these data do not show that apoptosis does not influence microglia, but rather only that apoptosis alone is insufficient to fully restore neuron-microglial communication. Indeed, in Fig 3A, Mtz seems to have induced some increase in *mpeg1* expressing cells in *Bax* mutants. In contrast, the data on nucleoside release altering microglia clearly show that apoptotic cells do affect the microglial population. A more persuasive conclusion, consistent with all of the data, would be that BAX-dependent processes in neuronal cells influence microglia through both apoptotic and non-apoptotic mechanisms that are molecularly distinct. The authors should consider whether this explicit phrasing is acceptable and adds clarity.

Answer: We deeply appreciate your kind comments and suggestion that “BAX-dependent processes in neuronal cells influence microglia through both apoptotic and non-apoptotic mechanisms”. This is just the point that we wanted to delivery in the present study. We are quite sorry for the unclear writing on this strength in the previous manuscript. We tried our best to clarify the conclusions in the revised narration.

Specific text issues (not including grammatical and word choice issues which are too numerous to list):

The text should briefly explain what is meant by *cq55* mutant.

Answer: We are grateful to the careful estimation. To investigate novel factors in microglial development, we carried out a large scale forward genetic screen in an ENU mutagenesis zebrafish library. A total of

203 crossing pairs were screened. Some embryos from the 55th cross presented a notable loss of *apoeb*⁺ signals and was investigated in this manuscript. We named this candidate mutant as *cq55*, because ZIRC has assigned a line designation to every registered zebrafish laboratory. The designation of my laboratory is "cq", an abbreviation of "Chongqing" of our city. Therefore, "*cq55*" has no special meaning but the line name registered in ZIRC and ZFIN. The related description was added in the revised manuscript (page 6, 23, line 124-125, 506-509, highlighted by the orange underline).

The text should briefly explain the NTR-metronidazole system.

Answer: The NTR/MTZ system is widely utilized to ablate targeted cell in zebrafish. The bacterial Nitroductase (NTR)-mediated conversion of a nontoxic prodrug metronidazole (MTZ) into a cytotoxic agent is principally used in this system. Upon binding, MTZ is reduced by NTR, which is converted into a potent DNA interstrand cross-linking agent, eventually leading to cell death^{16,17}. In the present study, we generated a *Tg(NBT:DenNTR)* line, in which the fluorescent protein Dendra2 was fused to NTR and was driven by the NBT promoter to make it visible in the neurons. The related description was added in the revised manuscript (page 8, 28, line 169-173, 679-681, marked by **turquoise background**).

The abbreviation AO⁺ is not defined before it is used.

Answer: We apologized for this. AO is an abbreviation of Acridine Orange (AO). It is a cell-permeable, nucleic acid intercalating dye that emitted green fluorescence. It is utilized as an indicator of dead cells¹⁵. The related information of "AO" was added to the revised figure legends and manuscript accordingly.

Line 257 the sentence "Uncaged IP3 or glutamate was injected into *bax*^{*cq55*} midbrains" should read "Cage IP3 or glutamate was injected into *bax*^{*cq55*} midbrains"

Answer: We are quite sorry for the mistakes. It was corrected in the revised manuscript.

Fig 1K- The authors report rich factor, which is understood to be the fraction of genes in a GO pathway that are differentially expressed in the sample. Is it correct that 100% of the genes in the isocitrate dehydrogenase pathway were differentially expressed?

Answer: We gratefully appreciate your valuable suggestion. The rich factor in the present report indicated the fraction of genes in a GO pathway that were differentially expressed in the samples. We used the DAVID Bioinformatics Resources to enrich the GO pathway by referring to the differentially expressed genes. However, all of the genes involved in the isocitrate dehydrogenase pathway and microglia differentiation pathway that were included in DAVID Bioinformatics Resources were differentially expressed. To display the results of gene enrichment better, we replaced the rich factor with the count of differentially expressed genes in a GO pathway. The image data and corresponding information were included in the revised Supplementary figure legend (Supplementary Fig. 4j, supplementary page 6, line 88-90, marked **yellow background**).

In Fig 3A, MTZ seems to increase *mpeg1*⁺ cells in the Bax mutant. Please provide statistical analysis and if *mpeg1*⁺ cells are increased, discuss the divergence of *mpeg1* and *apoe* data. Also, should *mpeg1* be analyzed in other mutants, such as *dll* KO?

Answer: We appreciate the careful assessment and acknowledge the instrumental comments. The number of *mpeg1*⁺ cells was counted and presented in the revised Fig. 3b. The statistical results indicated that the counts of *mpeg1*⁺ cells increased significantly in the MTZ-treated *bax*^{cq55} mutants relative to the DMSO-treated controls, supporting that macrophages are responsive scavengers to apoptotic bodies and dying neurons^{15, 18}. However, no simultaneous appearance of *apoeb*⁺ signals in the MTZ-treated *bax*^{cq55} mutants. These data collectively suggested that neuronal apoptosis was required for the attraction of *mpeg1*⁺ cells which, however, failed to differentiate to the *apoeb*⁺ microglia in the *bax*^{cq55} mutants. The number of *mpeg1*⁺ cells was calculated in 3 dpf *dlb*^{-/-} midbrains accordingly, which was comparable between the siblings and *dlb*^{-/-} mutants (Fig. 4i, Supplementary Fig. 5k). However, quite limited *apoeb*⁺ and NR⁺ signals were observed in the *dlb*^{-/-} midbrains at the same time (Fig. 4h, i). The related description was included in page 8, 9, and 12 (line 174-176, 181-183, 265-267, marked by green underline).

Reviewer #3 (Remarks to the Author):

In this manuscript, Zhao et al identify neuronal Delta/Dlb as a novel signal that regulates microglia differentiation. The authors rigorously show that Delta is induced downstream of neuronal activity in a Bax/CAMKII/CREB-dependent manner. Thus, the authors uncouple the role of Bax in promoting neuronal apoptosis— which initially attracts microglia to the nervous system— from a newly identified role in cell signaling, which promotes microglia maturation after their arrival. This is an impressive body of work with impactful findings, since the brain-specific cues that specify and maintain microglia identity are currently not well understood. However, this manuscript requires extensive editing for grammar and syntax as it is currently difficult to understand, making the findings more challenging to appreciate. Following a few revisions (listed below), this study should be published with no delay.

Major Comments

1. The authors conclude that neuronal Delta signals directly through microglial Notch to regulate microglial development. However, direct evidence that Delta is acting through Notch on microglia is not provided. The authors should perform epistasis experiments to support this claim— for example:

- Inhibit microglial Notch while rescuing neuronal delta in the Bax mutant, or
- Inhibit microglial Notch while uncaging Ca^{2+}

Answer: We are grateful to the careful estimation and constructive advices. To address this issue, we specifically disrupted the Notch activity in the zebrafish microglia though generating the *Tg(corola:DN-MAML-FLAG)* strain, in which a dominant-negative (DN) isoform of the murine mastermind-like (MAML) protein was fused with FLAG and controlled by the myeloid-specific *corola* promoter. We supplied neuronal Delta by transiently providing *HuC:dlb* in the *bax^{ca55}* mutants. We found that synergistic supplementation of *dlb* in *HuC⁺* cells and *DN-MAML-FLAG* in *corola⁺* cells in 3 dpf *bax^{ca55}* mutants notably reduced the recovery of *apoeb⁺* and *NR⁺* cells to approximately 13.64%, from 29.04% of the larvae supplying *dlb* (Fig. 4k, l). The information was added in the revised text (page 12-13, line 268-272, marked by grey background).

2. Line 63-65: The authors state that maturing microglia begin to engulf dying cells. However, none of the figures presented demonstrate the engulfment of “gradually appeared Casp3⁺ apoptotic cells.” The authors should provide images and 3D reconstruction showing this result.

Answer: We deeply appreciate the careful estimation. We carried out a time-lapse imaging on *Tg(corola:DsRed; ubiq:secAnnexinV-mVenus)* embryos^{19, 20} from 52 hpf, in which the brain microglia and dying cells are marked by DsRed and mVenus, respectively. The data indicated that pre-microglia gradually phagocytosed AnnexinV-labeled fragments of cell debris and manifested typical amoeboid morphology. 3D rendering was conducted on the *corola-DsRed⁺*, *apoeb-GFP⁺*, and active *Casp3⁺* signals. The result indicated that active *Casp3⁺* signals were engulfed by *apoeb-GFP⁺* cells which became amoeboid gradually. The data were included in the revised Supplementary Fig. 1c-d. The description was added accordingly (page 5, line 105-107, marked by red underline).

3. The manuscript requires extensive editing for grammar and syntax, including the abstract. For example:

- Line 22-24: “Bax concomitantly retains the pre-microglia through apoptotic neurons to ensure effective obtain of Notch instructions from the surrounding neurons. Notch signalling in regulating microglia signatures is faithful.” The first sentence should be rewritten, and the word “faithful” should be changed to “conserved.”

- Line 32: “Accumulating evidences indicate”

- Line 40: “It was recently suggested that rescue of damaged microglia or generation of safe microglia surrogates from iPSCs could prevent even reverse neurological disease progression.” What does “safe microglia surrogates” mean? Healthy? Homeostatic?

- Use of “fortuitously” on line 120 and “unfortunately” on line 122 appears emotional and unnecessary.

- Line 315: “This result followed the failed rescued outcomes of solely increasing apoptotic cells in baxcq55 brain.” This sentence is confusing and should be written, perhaps: “This result is consistent with the observation that solely increasing apoptotic cells in baxcq55 brain failed to rescue the microglia differentiation phenotype.”

Answer: Thank you very much for your careful evaluation and kind suggestion. We apologize for the previous poor writing. We made a substantial revision in the manuscript. The improper introductions were removed and several sections were re-organized. We asked both the English language editing service and a native speaker to polish the language and grammar. The obvious mistakes were corrected accordingly (page 2, 6, 7, 9, line 24-26, 35-36, 127-131, 193-194, marked by blue underline).

4. The authors should carefully check the relevance of the literature cited to assess whether the findings in these papers actually support their arguments. Some examples where papers were inappropriately cited include:

- Line 35-36: The authors state that NGF is a neurotrophic factor that supports the proliferation of neural progenitor cells, but cite a paper that demonstrate NGF’s role as a killing agent (Frade et. al, 1998).

- Line 37-38: The authors cited a paper that demonstrated microglial functions in mediating forgetting and erasure of memory (Wang et. al, 2020), but the authors used this reference to support an argument that microglia improve learning and memory performance.

- Line 112: The authors state that microglia heterogeneity has been demonstrated in mammals, but cite a paper focusing on differences between CNS border-associated macrophages and microglia (Utz et. al, 2020).

Answer: We deeply apologize for the unclear even incorrect citations. Several misquotes were carefully examined and amended in the revised manuscript (page 2, 3, 33, line 37-41, 808-818, marked by turquoise background). We appreciated the careful reviewing again!

Minor Comments

1. The colouring in Figure 1B is really confusing. It would be better to use only 2 colours, one for “sib” and one for “cq55.” The authors should also define the term “sib” in the text and/or legend.

Answer: We appreciate the kind suggestion. We used 2 colors to simplify sib and cq55 respectively in the revised Fig. 1c. The “sib” was defined in the revised legend accordingly (page 40, line 1054, marked by grey background).

2. Figure 1H should be presented as an earlier panel, as this information can help readers better understand the labeling used for each cell type.

Answer: Many thanks to the kind suggestion. It was done.

3. Could the increased contact between microglia and neurons in Figures 1E and 1F be explained by increased neuronal density during development? This factor should be considered and discussed.

Answer: We appreciated the careful assessment. The regarding information was included in the revised discussion (page 20, line 435-439, marked by blue color).

4. On line 125, the authors state that “More than half of the cells entered into the eyes, hindbrains, and ventral brains, which caused a drastic reduction in c-D⁺ pools,” but Figure 1I-J do not show data supporting this claim.

Answer: We are quite sorry for the mistakes of figures referring in the previous Figure 1i-j. The corresponding figure for this sentence should be previous Fig. 1I-m. We corrected the mistakes in the revised manuscript (page 7, line 134-136, marked by green underline).

5. Figure 1L-N are not referenced or explained in the main text. Please describe this experiment. It is unclear what Figure S1F is showing.

Answer: We deeply apologize for the incomplete reference and explanation of Fig. 1I-n and unclear description of Supplementary Fig. 1f in the previous manuscript. (a) Previous Fig. 1I-n showed that in a newly identified *cq55* mutant, comparable c-D⁺ cells initially appeared in the 52 hpf optic tectum; however, they did not remain there but mainly (more than 50%) exited to the eyes, hindbrain (HB), and ventral brain (VB), which caused a drastic reduction of c-D⁺ pools at 4 dpf. (b) In zebrafish, *apolipoprotein E (apoeb)*, a neurotrophic lipid carrier, is a well-known marker of microglia^{1, 21}. We validated that *apoeb* co-localize with microglia-specific markers *p2ry12* and *p2ry6* at 3 dpf in previous Supplementary Fig. 1f. We included the regarding explanation of Fig. 1I-n and Supplementary Fig. 1f in the revised text (Fig. 1i, j, Supplementary Fig. 1i, page 6, 7, line 119-120, 134-136, marked by green underline). Many thanks again!

6. On line 174-180, the authors list GO terms that are altered in mutants. For clarity, the authors should explicitly state the GO terms enriched for downregulated versus upregulated genes. This should also be made clear in Figure 3C.

Answer: We appreciate the careful assessment and apologize for lacking the explicit description of the GO terms in the downregulated versus upregulated genes. The KEGG pathways of enriched down-regulated genes were indicated in Fig. 3d. The corresponding information were included in the revised manuscript and figure legend (page 44, line 1091-1092, marked by yellow background).

7. Figure 1N should have more neurons drawn in the mutant condition.

Answer: We appreciated the careful assessment. The revised schematic diagram was presented in Fig. 1k.

8. Figure S8G should be included in the main figures.

Answer: Many thanks for your valuable advices. It was done.

9. Please introduce acronyms and field-specific terminology when first using them, including in figure legends. Some examples of terms that should be defined are:

- NR

- WISH

- *Z-bax*, *M-bax*, *H-bax*

Answer: We appreciated the careful assessment and apologized for the incomplete description of acronyms and field-specific terminology. NR is an abbreviation of “neural red”. WISH is an abbreviation of “whole mount in situ hybridization”. *Z-bax*, *M-bax*, and *H-bax* are abbreviations of zebrafish-, mouse-, and human-*bax*, respectively. The information of these acronyms was included in the revised figure legends.

10. How specific is *apoeb* as a marker for microglia maturation? Since *apoeb* is the marker used as the main readout of this study, further background or support for using this marker should be provided.

Answer: We appreciate the careful assessment. *apoeb* is a neurotrophic lipid carrier. It is a microglia marker widely used in zebrafish^{1,21}. During zebrafish development, embryonic macrophages invade the brain and colonize the optic tectum to become pre-microglia at about 60 hpf. These pre-microglia undergo phenotypic transformation and gradually turn on the expression of *apoeb*. *apoeb* is highly enriched in microglia and less abundant in other CNS cells²². The *apoeb*⁺ cells exhibited enhanced phagocytosis of neutral red particles and accepted to be matured microglia at embryonic stages¹. The background was added accordingly (page 4, line 66-68, marked by blue underline).

11. Figure 7M is a great schematic but should not be labeled “mouse.” The authors rigorously demonstrate the role of the schematized pathway in zebrafish, but the mouse data presented is not sufficient to conclude that this pathway as schematized regulates microglia maturation in mice.

Answer: Many thanks to your careful evaluation and suggestions. The schematic diagram was corrected accordingly.

12. On line 155, the authors state that “Bax is an indispensable neuronal factor in training microglia.” The word “training” should be changed, perhaps to “regulating microglia maturation,” since immune cell training describes a specific, separate biological phenomenon similar to microglial priming.

Answer: Many thanks to the kind suggestion and the description was revised accordingly (page 8, line 161-162, marked by grey background).

References

1. Herbomel P, Thisse B, Thisse C. Zebrafish early macrophages colonize cephalic mesenchyme and developing brain, retina, and epidermis through a M-CSF receptor-dependent invasive process. *Dev Biol* **238**, 274-288 (2001).
2. Li Y, Du XF, Liu CS, Wen ZL, Du JL. Reciprocal regulation between resting microglial dynamics and neuronal activity in vivo. *Dev Cell* **23**, 1189-1202 (2012).

3. Mazzolini J, *et al.* Gene expression profiling reveals a conserved microglia signature in larval zebrafish. *Glia* **68**, 298-315 (2020).
4. Ginhoux F, Prinz M. Origin of microglia: current concepts and past controversies. *Cold Spring Harb Perspect Biol* **7**, a020537 (2015).
5. Matcovitch-Natan O, *et al.* Microglia development follows a stepwise program to regulate brain homeostasis. *Science* **353**, aad8670 (2016).
6. Wong K, *et al.* Mice deficient in NRROS show abnormal microglial development and neurological disorders. *Nat Immunol* **18**, 633-641 (2017).
7. Nazmi A, *et al.* Chronic neurodegeneration induces type I interferon synthesis via STING, shaping microglial phenotype and accelerating disease progression. *Glia* **67**, 1254-1276 (2019).
8. Mullins MC, Hammerschmidt M, Haffter P, Nüsslein-Volhard C. Large-scale mutagenesis in the zebrafish: in search of genes controlling development in a vertebrate. *Curr Biol* **4**, 189-202 (1994).
9. Xu J, *et al.* Temporal-Spatial Resolution Fate Mapping Reveals Distinct Origins for Embryonic and Adult Microglia in Zebrafish. *Dev Cell* **34**, 632-641 (2015).
10. Li L, Jin H, Xu J, Shi Y, Wen Z. Irf8 regulates macrophage versus neutrophil fate during zebrafish primitive myelopoiesis. *Blood* **117**, 1359-1369 (2011).
11. Anderson SR, *et al.* Developmental Apoptosis Promotes a Disease-Related Gene Signature and Independence from CSF1R Signaling in Retinal Microglia. *Cell Rep* **27**, 2002-2013.e2005 (2019).
12. Wei Z, Chigurupati S, Arumugam TV, Jo DG, Li H, Chan SL. Notch activation enhances the microglia-mediated inflammatory response associated with focal cerebral ischemia. *Stroke* **42**, 2589-2594 (2011).
13. Cheng Z, *et al.* Inhibition of Notch1 Signaling Alleviates Endotoxin-Induced Inflammation Through Modulating Retinal Microglia Polarization. *Front Immunol* **10**, 389 (2019).
14. Grandbarbe L, Michelucci A, Heurtaux T, Hemmer K, Morga E, Heuschling P. Notch signaling modulates the activation of microglial cells. *Glia* **55**, 1519-1530 (2007).
15. Casano AM, Albert M, Peri F. Developmental Apoptosis Mediates Entry and Positioning of

- Microglia in the Zebrafish Brain. *Cell Rep* **16**, 897-906 (2016).
16. Curado S, Anderson RM, Jungblut B, Mumm J, Schroeter E, Stainier DY. Conditional targeted cell ablation in zebrafish: a new tool for regeneration studies. *Dev Dyn* **236**, 1025-1035 (2007).
 17. Curado S, Stainier DY, Anderson RM. Nitroreductase-mediated cell/tissue ablation in zebrafish: a spatially and temporally controlled ablation method with applications in developmental and regeneration studies. *Nat Protoc* **3**, 948-954 (2008).
 18. Sieger D, Moritz C, Ziegenhals T, Prykhozhij S, Peri F. Long-range Ca^{2+} waves transmit brain-damage signals to microglia. *Dev Cell* **22**, 1138-1148 (2012).
 19. Li L, Yan B, Shi YQ, Zhang WQ, Wen ZL. Live imaging reveals differing roles of macrophages and neutrophils during zebrafish tail fin regeneration. *J Biol Chem* **287**, 25353-25360 (2012).
 20. Morsch M, *et al.* In vivo characterization of microglial engulfment of dying neurons in the zebrafish spinal cord. *Front Cell Neurosci* **9**, 321 (2015).
 21. Peri F, Nüsslein-Volhard C. Live imaging of neuronal degradation by microglia reveals a role for v0-ATPase a1 in phagosomal fusion in vivo. *Cell* **133**, 916-927 (2008).
 22. Thiel WA, Esposito EJ, Findley AP, Blume ZI, Mitchell DM. Modulation of retinoid-X-receptors differentially regulates expression of apolipoprotein genes *apoc1* and *apoeb* by zebrafish microglia. *Biol Open* **11**, (2022).

REVIEWER COMMENTS

Reviewer #1 (Remarks to the Author):

The authors carefully and with high detail responded to this reviewer's comments and appear to have done so more broadly, making the paper much stronger. I think there are some mild over-statements about the meaning of the mouse experiments and other points of interpretation or clarification, that I have noted below, with the goal to increase clarity and make sure statements match data, but they are quite minor. Overall I think this is an interesting story that will be appreciated by the field and has an impressive level of validation.

Fig 8d - the legend is either mislabeled or confusing formatting (the rbpj fl/fl goes with the green dots, but looks like it could also be the orange dots)

Fig 8g - not clear what the "in 1 slice" adds if the unit of mm² is already specified.

Lines 45-7 - edit sentence - seems like authors are implying microglia enter the mouse brain at e13.5-14.5, but they are thought to enter earlier (depending on who you ask/how you count, ~ E9)

Line 102 - what's the units for the number 3 (3 per field, per brain?) - same for line 107

Line 136 - consider adjusting this to something along lines of "these data indicate compromised microglial recruitment and/or maturation in cq55 embryos" - eg, the data shown at that point does not establish that it's purely a maturation deficit, it could also relate to recruitment?

Lines 370-2 - clarify the interpretation here about dll3 being likely absent in bax ko brain - by western it does not seem absent?

379 - the title is overstated for this section (maybe change eliminates to impairs or disrupts)

Line 403 - I dont think the data shows that notch signaling sustains the microglia signature (signature genes are not expressed at high levels, just at mildly higher levels than mcsf alone)

Line 415 - suggest toning down or editing sentence to clarify what results the authors are referring to - the in vitro results do not really address whether notch is indispensable for triggering embryonic microglia differentiation in mice, they show that dll3 ligand in vitro has a very modest effect on microglial signature gene expression, but perhaps the authors are summarizing all of the mouse studies.

442-5 - overstated - the in vitro expts did not assess microglial maturation, just expression of some signature genes.

Reviewer #2 (Remarks to the Author):

The authors have thoughtfully addressed all of the concerns raised in my review. The manuscript is excellent and informative and is ready for publication.

Reviewer #3 (Remarks to the Author):

The authors have addressed all my concerns.

Reviewer #4 (Remarks to the Author):

This is an impressive body of work and is important for the field. The field (partially discovered by these authors previously) has shown that cell death of neurons is a component of microglia colonization. However, the mechanism of that concept has not been thoroughly understood. Here, the authors show an impressive set of experiments to present a new pathway in microglia differentiation (or maturation). The zebrafish experiments are excellent and could be published on their own. The inclusion of the mouse experiments are a nice addition but the zebrafish work could stand on its own. The work should be published in its current form.

REVIEWERS' COMMENTS

Reviewer #1 (Remarks to the Author):

The authors carefully and with high detail responded to this reviewer's comments and appear to have done so more broadly, making the paper much stronger. I think there are some mild over-statements about the meaning of the mouse experiments and other points of interpretation or clarification, that I have noted below, with the goal to increase clarity and make sure statements match data, but they are quite minor. Overall I think this is an interesting story that will be appreciated by the field and has an impressive level of validation.

Fig 8d - the legend is either mislabeled or confusing formatting (the rbpj fl/fl goes with the green dots, but looks like it could also be the orange dots)

Answer: Many thanks to your careful evaluation and suggestions. We apologize for the confusion due to the mistakes of symbols in the figure. The error has been corrected in the revised Fig. 8d.

Fig 8g - not clear what the "in 1 slice" adds if the unit of mm² is already specified.

Answer: We are sorry for the confused annotation. Actually, each dot represents an individual *in vitro* assay of the IBA1⁺ cell density per slice. We revised Fig. 8g to make it clear. Thanks again for your comments.

Lines 45-7 - edit sentence - seems like authors are implying microglia enter the mouse brain at e13.5-14.5, but they are thought to enter earlier (depending on who you ask/how you count, ~ E9)

Answer: Thank you very much for your kind comments. We apologize for the confused information. The description was revised accordingly (page 3, line 46-48, marked by **yellow background**).

Line 102 - what's the units for the number 3 (3 per field, per brain?) - same for line 107

Answer: We apologize for the unclear description. The units for the number 3 and 25 are "per midbrain". The explanation was included in the revised manuscript (page 5, line 103-104, 109, marked by **gray background**). Many thanks again!

Line 136 - consider adjusting this to something along lines of "these data indicate compromised microglial recruitment and/or maturation in *cq55* embryos" - eg, the data shown at that point does not establish that it's purely a maturation deficit, it could also relate to recruitment?

Answer: We gratefully appreciate the careful assessment and kind suggestion. The description was revised accordingly (page 7, line 138-139, marked by **yellow background**).

Lines 370-2 - clarify the interpretation here about dll3 being likely absent in bax ko brain - by western it does not seem absent?

Answer: Many thanks to your careful evaluation and suggestions. We are sorry for the improper

interpretation about dll3. The description was revised accordingly (page 17, line 379-381, marked by green background).

379 - the title is overstated for this section (maybe change eliminates to impairs or disrupts)

Answer: Many thanks to the kind suggestion and the description was revised (page 18, line 390, marked by turquoise background).

Line 403 - I don't think the data shows that notch signaling sustains the microglia signature (signature genes are not expressed at high levels, just at mildly higher levels than mcsf alone)

Answer: We appreciated the careful assessment. The description was corrected in the revised manuscript (page 19, line 415, marked by gray background). Many thanks again!

Line 415 - suggest toning down or editing sentence to clarify what results the authors are referring to - the in vitro results do not really address whether notch is indispensable for triggering embryonic microglia differentiation in mice, they show that dll3 ligand in vitro has a very modest effect on microglial signature gene expression, but perhaps the authors are summarizing all of the mouse studies.

Answer: Many thanks to the kind suggestion and the description was revised accordingly (page 19, line 427-429, marked by green background).

442-5 - overstated - the in vitro expts did not assess microglial maturation, just expression of some signature genes.

Answer: We apologize for the overstatement. The revised description was included in page 21, line 456-458, marked by yellow background. We appreciated your great efforts again!

Reviewer #2 (Remarks to the Author):

The authors have thoughtfully addressed all of the concerns raised in my review. The manuscript is excellent and informative and is ready for publication.

Answer: We sincerely appreciate your comments.

Reviewer #3 (Remarks to the Author):

The authors have addressed all my concerns.

Answer: Thanks very much.

Reviewer #4 (Remarks to the Author):

This is an impressive body of work and is important for the field. The field (partially discovered by these authors previously) has shown that cell death of neurons is a component of microglia colonization. However, the mechanism of that concept has not been thoroughly understood. Here, the authors show an impressive set of experiments to present a new pathway in microglia differentiation (or maturation). The zebrafish experiments are excellent and could be published on their own. The inclusion of the mouse experiments are a nice addition but the zebrafish work could stand on its own. The work should be published in its current form.

Answer: We are grateful to your careful assessment and acknowledge the comments.